# Comparative study between ground-based observations and NAVGEM-HA analysis data in the MLT region

Gunter Stober[1,2], Kathrin Baumgarten[2,3], John P. McCormack[4], Peter Brown[5,6], and Jerry Czarnecki[2]

[1]Institute of Applied Physics, Microwave Physics, University of Bern, Bern, Switzerland
[2]Leibniz-Institute of Atmospheric Physics at the University of Rostock, Kühlungsborn, Germany
[3]Fraunhofer Institute for Computer Graphics Research IGD, Rostock, Germany
[4]Space Science Division, Naval Research Laboratory, Washington DC
[5]Dept. of Physics and Astronomy, University of Western Ontario, London, Ontario, Canada N6A 3K7
[6]Western Institute for Earth and Space Exploration, University of Western Ontario, London, Ontario, N6A 5B7, Canada

**Correspondence:** gunter.stober@iap.unibe.ch

**Abstract.** Recent studies have shown that day-to-day variability of the migrating semidiurnal (SW2) solar tide within the mesosphere and lower thermosphere (MLT) is a key driver of anomalies in the thermosphere-ionosphere system. Here we study the variability in both the amplitude and phase of SW2 using meteor radar wind and lidar temperature observations at altitudes of 75-110 km as well as wind and temperature output from NAVGEM-HA, a high-altitude meteorological analysis system. Application of a new adaptive spectral filter technique to both local radar wind observations and global NAVGEM-HA analyses offers an important cross-validation of both data sets and makes it possible to distinguish between migrating and non-migrating tidal components, which is difficult using local measurements alone. Comparisons of NAVGEM-HA, meteor radar, and lidar observations over a 12-month period show that the meteorological analyses consistently reproduces the seasonal as well as day-to-day variability in mean winds, mean temperatures, and SW2 features from the ground-based observations. This study also examines in detail the day-to-day variability in SW2 during two sudden stratospheric warming, events that have been implicated in producing ionospheric anomalies. During this period, both meteor radar and NAVGEM-HA winds show a significant phase shift and amplitude modulation, but no signs of coupling to the lunar tide as previous studies have suggested. Overall, these findings demonstrate the benefit of combining global high altitude meteorological analyses with ground-based observations of the MLT region to better understand the tidal variability in the atmosphere.

## 1 Introduction

There is a growing need to understand the global wind field from the surface up to the lower thermosphere (0-100 km) and beyond as well as its day-to-day variability due to meteorological processes. Planetary waves and atmospheric tides are dominant drivers at the mesosphere and lower thermosphere (MLT) that provide a highly variable dynamical lower boundary to the thermospheric/ionospheric system, e.g. at the equatorial dynamo region at altitudes from 100 to 150 km (see, e.g., Akmaev,

2011, and references therein). The upward propagation of these drivers from their source regions near the surface into the MLT region is determined in large part by the global wind field. Accurate assessments of both daily and seasonal variability in winds and tidal modes has therefore become necessary for better understanding lower atmospheric forcing of the thermosphere/ionosphere system.

At mid- and polar latitudes planetary waves provide a significant contribution to the variability of the winter MLT and play a major role in vertical coupling processes between the different atmospheric layers. For example, during sudden stratospheric warmings (SSWs) (Matsuno, 1971; Andrews et al., 1987) the whole middle atmosphere (stratosphere/mesosphere) responds to sudden reversals of the zonal wind from eastward to westward and back to eastward accompanied by an increase of the strato-
spheric temperature and a mesospheric cooling (see, e.g. Chandran et al., 2014; Zülicke et al., 2018, , and references therin). SSWs are often studied using General Circulation Models (GCMs), which are either free-running (e.g., GAIA, WACCM, KMCM, Jin et al. (2012); Liu et al. (2010); Becker (2017); Zülicke et al. (2018)) or nudged to reanalysis fields (e.g., SD-WACCM, Marsh (2011); Stray et al. (2015); Limpasuvan et al. (2016)). Manney et al. (2008, 2009) characterized the SSW in 2006 as a vortex displacement and the SSW in 2009 as a vortex splitting event making use of global satellite observations
(MLS-Microwave Limb Sounder) and data assimilated reanalysis mostly at the stratosphere and lower mesosphere. Matthias et al. (2013) investigated the role of planetary waves in the evolution of vortex splitting and displacement events combining satellite data and ground-based observations.

Atmospheric tides are generated in the troposphere and stratosphere mostly through the absorption of sunlight by water vapor and ozone (e.g., Lindzen, 1979). They have been studied theoretically (e.g., Chapman and Lindzen, 1970; Forbes, 1982; Wang
et al., 2016) and from observations (e.g., Portnyagin et al., 1993; Merzlyakov et al., 2009; Oberheide et al., 2009, 2011, and references therein) for decades. More recent studies analyzed the response of the semidiurnal tide during SSWs using ground-based instruments and nudged GCM data or investigated the relative importance and impact of the semidiurnal lunar tide during SSWs with TIME-GCM and WACCM (Pedatella et al., 2012; Pedatella and Maute, 2015). However, atmospheric tides propagate from their source region up to the MLT through a constantly varying altitude dependent wind and temperature field,
which significantly modifies the phase of the tides, depending on their vertical wavelength, as well as the vertical wavelength itself.

In this study, we compare local meteor radar (MR) wind observations as well as lidar temperature measurements with meteorological analyses produced with NAVGEM-HA (Navy Global Environmental Model - High Altitude), a data assimilation and modeling system that extends from the surface to the lower thermosphere. NAVGEM-HA fields were available from December
2009 to December 2010 and during the winter season 2012/13 starting in December 2012 until March 2013. Recent studies (Eckermann et al., 2018; McCormack et al., 2017) have presented initial cross-validation of the mesospheric winds from NAVGEM-HA for two winter seasons using worldwide distributed MR measurements. Here, we extend these initial comparisons to include seasonal mean winds (30-day median) from NAVGEM-HA and from three MRs at mid- to high latitudes for the year 2010. Time series of both NAVGEM-HA analysed winds and MR measurements are decomposed into daily mean
winds, tides and GW residuals using a recently introduced analysis technique called adaptive spectral filter (ASF) (Stober et al.,

2017; Pokhotelov et al., 2018; Wilhelm et al., 2019; Baumgarten and Stober, 2019). This technique is designed to extract daily mean winds and tidal variations on a day-to-day basis. In addition to MR measurements, we also present the first comparison between midlatitude temperature observations from a resonance lidar and NAVGEM-HA analysed temperatures for the 2010 period.

Meteorological analysis data, such as NAVGEM-HA, provide a much more realistic forcing of the upper atmosphere due to tides and mean winds compared to current versions of other comprehensive models. Chandran and Collins (2014) investigated SSW events using WACCM-SD nudged with reanalysis fields from the GEOS-5.2 reanalysis system up to an altitude of about 40 km. However, at altitudes above 70-80 km the nudged model started to substantially deviate from the observed wind climatologies (Wilhelm et al., 2019). In particular, the nudged model showed a wind reversal from eastward to westwards winds

between 70-80 km, which is not confirmed from the wind climatologies. Such reversal of the zonal wind can be also found in other comprehensive models or mechanistic models (Smith, 2012; Becker, 2012). Liu (2016) shows a comparison among several GCMs indicating that there are substantial deviations at the mesosphere and upper atmosphere, although each of the GCMs was nudged up to the lower stratosphere (see also Pedatella et al. (2014) for more details). Only the GAIA model (Jin et al., 2012; Liu et al., 2014) showed during winter eastward winds at the MLT. Previously, the eCMAM model was also cross-

validated with ground-based meteor radar observations to investigate mean winds and tides and their amplitude and phase behavior at equatorial latitudes (Du et al., 2007; Ward et al., 2010). They found a remarkably good agreement between the model and the local observations using 60-day running means underlining the value of such comparisons. The focus of the present study is to examine the degree of agreement between day-to-day and as well as seasonal variability in SW2 between a global meteorological analyses of the MLT region and ground-based observations as a means to, ultimately, better understand

the origins of this variability.

Finally, we perform a detailed comparison of SW2 variability from both NAVGEM-HA and meteor radar observations during the SSWs in 2009/10 and 2012/13, focusing in particular on how the amplitude and phase of semidiurnal variability in both data sets responds to changes in the background wind. Overall, the results of these comparisons show very good agreement between NAVGEM-HA analysed winds and MR observations, highlighting the utility of combining global high altitude data

assimilation products with ground-based observations of the MLT to lend new insight into the causes of semidiurnal tidal variability over daily to seasonal time scales. Such short time variations are essential for the understanding of the forcing from below of the thermosphere and ionosphere (Liu, 2016). Therefore, the paper is structured as follows. First, we describe the observations for winds and temperatures in the MLT region and the corresponding meteorological analysis data in Section 2. Section 3 provides a detailed explanation of the methodology used for the data analysis. Section 4 presents the results for the

climatology, comparing mean winds simultaneously seen in the meteor radar data at different locations with the NAVGEM-HA analysis data accompanied with available temperature measurements from a resonance lidar at one mid-latitude location. The results are also discussed for the semidiurnal tide for the whole year (2010) as well as during the winter season in 2010 and 2013 in Section 5. Finally, the findings are summarized and a conclusion is given in Section 6.

## 2 Data description

### 2.1 Wind observations

In this study, we compare the 3-hourly global synoptic wind and temperature analyses from NAVGEM-HA with meteor radar observations collected at three different latitudes in Andenes (69° N, 11° E) in Norway, Juliusruh (54.3° N, 13° E) in Germany and Tavistock (CMOR- Canadian Meteor Orbit Radar) (43.2° N, 80.7° W) in Canada. All three meteor radars use the same software for meteor detection and classification as described in Hocking et al. (2001). All systems were almost continuously in operation for the analyzed periods. Only the Andenes system shows some data gaps, mainly due to the more extreme weather conditions in Northern Norway, which caused some damage to the antennas and from time to time a power outage. A more detailed description of the CMOR radar can be found in Brown et al. (2008). A summary of the Juliusruh and Andenes MR is found in Stober et al. (2012) and Wilhelm et al. (2017).

MLT winds are obtained with a temporal resolution of 1 hour and a vertical resolution of 2 km using the wind retrieval algorithm presented in Stober et al. (2018), which is a further development of the wind analysis presented in Hocking et al. (2001). The wind analysis contains a full error propagation of the statistical uncertainties and a physical error model based on the vertical and temporal shear as spatio-temporal Laplace filter for each wind component. Contrary to many other meteor radar wind analysis, the algorithm also solves for the vertical wind velocity. The obtained mean vertical velocities show values of a few cm/s and are mainly used as quality control for successful convergence of the wind fit. In the present study, we use 4 meteors as a minimum for a successful wind fit.

### 2.2 Temperature observations

At Kühlungsborn (54° N, 12° E), around 118 km southwest of the meteor radar at Juliusruh, a resonance lidar was in operation until 2012 to observe temperatures in the MLT region. The potassium lidar measures the Doppler broadening of the 770 nm potassium D1 resonance line by scanning with a narrow band Alexandrite ring laser. The system is fully daylight capable. Further details can be found in von Zahn and Höffner (1996); Fricke-Begemann et al. (2002).

The extent of the potassium layer in the atmosphere limits the range of heights at which temperatures can be determined. In this work, temperatures are determined for heights between 80 and 105 km. The integration time of the data used here is 1 h with a shift of 15 min. The vertical resolution is 1 km. In addition to the resonance lidar, also a Rayleigh-Mie-Raman (RMR) lidar was operated during the night at the same location until 2013. This lidar used the second harmonic output of a Nd:YAG laser at 532 nm. The temperatures are calculated under the assumption of hydrostatic equilibrium from the Rayleigh backscatter which is proportional to the atmospheric air density (Hauchecorne and Chanin, 1980). The initial temperature value for integration is taken from the resonance lidar (Alpers et al., 2004). The temperatures from the RMR lidar cover an altitude range between 22 and 90 km. But as the focus of this study is on the MLT region, we use these temperatures only above 70 km.

Here, daily mean temperatures as a composite between 2003 and 2012 are used to describe the mean temperature field during the year in the MLT region. A full description of the seasonal variation has been published in Gerding et al. (2008).

## 2.3  NAVGEM-HA meteorological analyses

NAVGEM-HA is a high-altitude numerical weather prediction (NWP) system extending from the surface to ∼116 km altitude that provides atmospheric winds, temperatures and constituent information. It is based on the operational system described in Hogan et al. (2014), which combines the NAVGEM global spectral forecast model with a hybrid four-dimensional variational (4DVAR) data assimilation algorithm (Kuhl et al., 2013).

In addition to standard operational meteorological observations in the troposphere and stratosphere, NAVGEM-HA assim-
ilates satellite-based observations of temperature, ozone and water vapor in the stratosphere, mesosphere and lower thermo-sphere (McCormack et al., 2017). The NAVGEM-HA output is on a $1°$ latitude and longitude grid, respectively. The temporal resolution of the data output fields is 3 hours. NAVGEM-HA uses a fixed top level pressure of $6 \cdot 10^{-5}$ hPa (e.g., McCormack et al., 2017; Eckermann et al., 2018, and references therein), which corresponds to an approximate altitude of 116 km. How-ever, at the upper three model levels, an enhanced diffusion is applied to reduce the effects of wave reflection. These layers
effectively act as a "sponge layer" and are not included in the data analysis. The forecast model component of NAVGEM-HA incorporates the same implicit fourth-order horizontal diffusion of vorticity, divergence, and virtual potential temperature used in its predecessor system (NOGAPS-ALPHA) to suppress growth of unrealistic variances near the truncation scale, as described in McCormack et al. (2015). Default values for the diffusion result in an effective e-folding time of 24 hours at the highest wavenumber (here T119). In the top 3 model levels the diffusion is ramped up to produce an effective e-folding of  2
20  hours at the top level. In the 74-level version of NAVGEM-HA used in this study, this region of enhanced diffusion (sponge layer) covers levels with $p <$ 1.e-3 hPa or  95 km in pressure-altitude.

In an initial validation study, McCormack et al. (2017) used NAVGEM-HA output interpolated to geometric altitudes up to 95 km for the mean winds and up to 90 km for the wave analysis. In the present study, vertical profiles of NAVGEM-HA analyzed winds and temperatures are converted from the model vertical grid in geopotential altitude to a geometric altitude
grid as done in Eckermann et al. (2009) up to 94 km altitude. Above this level, NAVGEM-HA vertical resolution degrades significantly, as the vertical grid spacing increases from ∼3 km near 80 km altitude to more than 5 km near 100 km altitude. To date, NAVGEM-HA winds and tides up to 90 km altitude have been shown to be in good agreement with both ground-based MR observations, as reported in McCormack et al. (2017), Eckermann et al. (2018), and Laskar et al. (2019), and with independent satellite-based wind observations as reported in Dhadly et al. (2018). The present study extends these initial validation studies
to include, for the first time, validation with two independent ground-based data sets over a 12-month period.

In this study we use a fixed geometric altitude grid (based on the World Geodetic System 84 model) with a maximum al-titude of 94 km and 2 km vertical resolution at the MLT to match the meteor radar data. We convert the geopotential altitudes of NAVGEM-HA to geometric altitudes. However, we note that the geopotential altitude of the highest usable output level, neglecting the sponge layer effects noted above, has a geometric altitude between 92 to 89 km. As a consequence tidal am-

plitudes above 90 km altitude should not be considered as geophysical and are caused by the extrapolation to the geometric altitude grid and sponge layer effects. Further, the vertical constraint implemented in the ASF amplifies this effect even more. At mesospheric altitudes, NAVGEM-HA assimilates satellite measurements from TIMED and AURA satellites and radiances from the Defense Meteorological Satellite program (DMSP) (Eckermann et al., 2018). Systematic differences between the

meteorological analysis and the ground-based wind and temperature data herein may have different origins. There could be intrinsic differences due to the model physics leading to such deviations or the assimilated data itself may show some systematic differences in relation to the observations used for the comparison. Further, considering that the *true* state of the atmosphere of temperature and winds remains elusive, it is hard to determine which of the observational techniques provides a better representation of this *true* state. Thus, it is essential to assess some of the systematic differences, which can arise due to the

methodology employed for the comparison e.g., does applying different diagnostics or different spatio-temporal sampling of the instruments make a large difference. Validation and assessment of potential biases between the SABER temperatures and ground-based lidar measurements can be found in Xu et al. (2006); Dawkins et al. (2018). A cross comparison of the MLS and SABER temperatures is presented in Schwartz et al. (2008). A detailed description of how the data assimilation in NAVGEM-HA treats the temperature biases between both satellites is given in Eckermann et al. (2018).

Another important point affecting the comparison is the availability of the assimilated data. Above 90 km less satellite observations can be assimilated. Further, it has to be noted that the spatial coverage of the assimilated SABER temperatures varies due to the yaw cycle of the spacecraft, which changes every 60 days the observing geometry providing a variable latitudinal coverage. From 52° S to 52° N the satellite collects constant measurements whereas the higher latitudes depend on the yaw cycle and alternates between up to 82° S or 82° N latitudinal coverage. This yaw cycle pattern may affect the quality of

NAVGEM-HA analyses at Juliusruh and Andenes.

## 3   Local and global diagnostics

One of the challenges comparing different data sets is the use of a common diagnostic to ensure that all observations and the meteorological analysis data are treated in the same way. In particular, observational data can be more difficult to be analyzed due to data gaps or uneven temporal sampling. Atmospheric tidal and planetary wave amplitudes are often obtained

from Fourier based techniques (e.g., Stockwell et al., 1996; Torrence and Compo, 1998). In the case of unevenly sampled data Lomb-Scargle periodigrams are used (Lomb, 1976; Scargle, 1982), which provide a amplitude/power spectrum and a significance level, but without a phase information. For observational data, it is also very common to derive the tidal information of amplitude and phase with a least-square fit (Lima et al., 2007) or by a multiple regression analysis assuming, for instance, a circular polarization for the semidiurnal tide (Jacobi et al., 2008).

A commonly used approach to extract tides is a harmonic analysis:

$$u, v, T = u_0, v_0, T_0 + \sum_{n=1}^{3} a_n \sin(\frac{2\pi}{P_n} \cdot t) + b_n \cos(\frac{2\pi}{P_n} \cdot t) \ ; \tag{1}$$

here $u, v, T$ are the zonal, the meridional wind and the temperature, $a_n$ and $b_n$ are the tidal Fourier coefficients, $P_n = 24, 12, 8$ stands for the tidal periods in hours and $t$ is the time of the observation either in UTC or local time, whatever is preferred. Harmonic tidal analysis work well for time series of several days or months, but assumes a constant mean background wind, tidal amplitude, and phase for the selected period. Recent studies of mean winds and tides using meteor radar, lidar and satellite observations indicate that tides have a fairly intermittent amplitude and phase character (Stober et al., 2017; Baumgarten et al., 2018; Baumgarten and Stober, 2019; Dhadly et al., 2018).

The adaptive spectral filter (ASF) aims to be a simple and general diagnostic to decompose time series in 1-D (temporal filter) (Stober et al., 2017) or 2-D (temporal-spatial filter) (Baumgarten and Stober, 2019). The technique is based on least-squares and, hence, applicable to unevenly sampled data and no additional zero-padding needs to be applied for data gaps as long as sufficient observations are available in the remaining adapted time window. Another benefit of the least-squares implementation is given in the error propagation to the derived quantities through the covariance matrix. The term 'adaptive' in this context relates, similar to the wavelet technique, that the window length adapts to the number of wave cycles for each frequency component that is fitted. The MR and NAVGEM-HA time series are decomposed into daily mean winds, diurnal tide, semidiurnal tide, terdiurnal tide and gravity wave residuum using the ASF.

The ASF uses a sliding window and fits each tidal component applying a scaling factor of 1.3 accounting for the number of wave cycles and no de-trending is applied. The scaling factor determines the window length that is used for the fitting for each frequency component. Here we applied a window length of 31 hours for the diurnal tide, whereas the semidiurnal tide is determined using a 16 hour window and so forth for the terdiurnal tide. At first, the daily mean wind and the diurnal tidal (amplitude and phase) components are determined considering also a semidiurnal and terdiurnal tide. In the next step, the semidiurnal tide is fitted using a regularization by the previously determined daily mean wind and diurnal tide and adapting the window length. The same procedure is repeated for the terdiurnal tide respectively. Due to the short window length, the bandwidth for each tidal component is rather wide and may also include some gravity wave contributions. It turns out that just applying temporal filtering leads to some contamination of the obtained tidal amplitudes and phases due to inertia gravity waves with short (less than 10 km) vertical wavelengths (see appendix A). However, there are also some studies from polar latitudes using lidar and radar observations from McMurdo/Scott base (77.8° S, 166.7° E) and from Syowa Station (39.6° E, 69.0° S) indicating the presence of gravity waves with vertical wavelengths of 22-23 km (Chen et al., 2013) or periods close to the semidiurnal tide (Shibuya et al., 2017). However, Davis et al. (2013) has shown that the diurnal and semidiurnal tide typically has vertical wavelengths larger than 20 km. Hence, we constrain our tidal amplitudes and phases by assuming that the phase of the diurnal and semidiurnal tide only gradually changes with altitude using a 16 km vertical retrieval kernel. The mean winds are constrained by a 10 km vertical retrieval kernel to avoid issues during the summer wind reversal from westward winds to eastward winds.

We optimized these vertical wavelength values considering the results of previous studies using meteor radars investigating the vertical wavelengths of tides (Yu et al., 2013; Davis et al., 2013; Fritts et al., 2019). These earlier studies showed that the vertical wavelengths for most of the tidal modes are much larger than >25 km. Only Yu et al. (2013) found for some Hough modes vertical wavelengths shorter than <25 km. To avoid a potential contamination of shorter tidal wavelengths in our

vertical retrieval kernel, we did not implement a hard cut off vertical wavelength. Instead, we just constrain the smoothness of the vertical tidal phase within the averaging kernel and even allow a gradual change. An example of the ASF(2D) tidal fit compared to ASF(1D) is presented in appendix A1.

The vertical regularization constraint is an essential feature of the ASF compared to many other diagnostic techniques based on wavelet or Fourier methods. Previous studies based on lidar observations (e.g., Ehard et al., 2015; Baumgarten et al., 2017, and reference therin) already investigated how the potential gravity wave energy changes with the applied filtering. Temporal filters tend to underestimate inertia gravity waves due to their long periods combined with short vertical wavelengths, whereas vertical filters are designed to eliminate the tidal contribution due to their large vertical wavelengths. As a consequence, this filter underestimates gravity waves with comparatively large vertical wavelengths. The ASF is much less prone to such biases due to the combination of spatio-temporal information for the specific waves. Since NAVGEM-HA produces global wind and temperature fields, we can extract tides as global waves and separate migrating and non-migrating tidal modes. Migrating tides are the DW1 (diurnal westward wave number 1), SW2 (semidiurnal westward wave number 2) and TW3 (terdiurnal westward wave number 3); all other tidal modes are non-migrating tidal components (e.g., Forbes et al., 2008; Miyoshi et al., 2017, and references therein). The migrating and non-migrating tidal components are obtained using the following function;

$$u, v, T = u_0, v_0, T_0 + \sum_{s=-3}^{3} \sum_{i=1}^{3} \Big( a_{si} \cdot \sin(s \cdot \lambda - \frac{2\pi}{P_i} \cdot t) + b_{si} \cdot \cos(s \cdot \lambda - \frac{2\pi}{P_i} \cdot t) \Big) + further\ waves \quad , \tag{2}$$

where $s$ is the zonal wave number (negative eastward, positive westward), $\lambda$ denotes the longitude at a fixed latitude circle, $P_i$ are the periods of the diurnal, semidiurnal and terdiurnal tide and $a_{si}$ and $b_{si}$ are the Fourier coefficients for each wave number $s$ and period $P_i$. The zonal mean zonal and meridional wind and the zonal mean temperature are given by $u_0, v_0, T_0$. The function also includes longer period waves such as the quasi two day wave (QTDW) with wave number s=1,2,3 and stationary planetary waves with wave number s=1,2,3 (Baumgarten and Stober, 2019; Schranz et al., 2019).

Daily mean tides for all the components are obtained by using a 3-day window around a central day, which is sufficient to still see some day-to-day variability and to determine potential phase drifts of each tidal component. The global tidal phase for all tidal components is referenced to the Prime meridian (Greenwich). Although NAVGEM-HA provides validated wind and temperature products from ∼18 km up to ∼94 km altitude, we focus our comparison to the MLT region and mostly to the available MR observations. A detailed discussion of the QTDW or planetary waves is beyond the scope of this paper and we leave these for other studies.

## 4 Results

In the first two parts of the results, we show the mean state of the atmosphere during the year in the MLT region using winds and temperatures from observations and NAVGEM-HA data. Next, the seasonal variation of the semidiurnal tidal component derived with the adaptive spectral filter is presented for each location. In addition to this, the analysis is also done for two examples of sudden stratospheric warming in the winter 2009/2010 and 2012/2013 to determine how well the observed varia-

tions in the MLT winds correspond to the NAVGEM-HA analysis data as well as to determine the day-to-day variability of the semidiurnal tide.

## 4.1  Mean winds

Fig. 1 shows the time variation of the zonal and meridional winds at the three locations Andenes, Juliusruh and Tavistock, from hourly meteor radar observations (left column), and the corresponding 3-hourly NAVGEM analyzed winds (center column) for the same location and each latitude as zonal mean values (right column). Daily mean winds are calculated and small scale variations such as tides and gravity waves are removed by the adaptive spectral filter and planetary waves are effectively filtered using a 30-day running median. The climatologies are based on the same time periods for MR winds and NAVGEM-HA and include December 2009 until December 2010 with periodic boundary conditions. In general, there is a good agreement of the seasonal wind pattern between the meteor radar wind observations and the NAVGEM-HA data. At all three locations, the zonal wind observations show the typical eastward winds in winter and the prominent wind reversal in spring. In particular, the seasonal asymmetry of the spring transition as well as the the gradual change of the summer wind reversal altitude seen in the meteor radar winds is well-reproduced in the NAVGEM-HA analyses.

During summer, a strong transition between westward and eastward winds occurs between 80 and 90 km altitude. The transition height decreases from high to midlatitudes. Above 90 km altitude, the eastward jet reaches wind velocities of about 40 m/s for all stations. The meridional winds during winter are typically northward, while they are southward during the summer. Similar behavior is seen in the NAVGEM-HA analysis data, but here the magnitude of the winds is to some extent larger compared to the meteor radar observations. Although the general morphology of the seasonal pattern is well captured in NAVGEM-HA, there are some differences in the wind reversal altitudes in summer in both wind components, which would affect the gravity wave breaking altitudes and, hence, the altitude of the resulting momentum deposition.

Furthermore, the altitude where the zonal wind reverses during summer decreases not as much with latitude as indicated from the meteor radar observations for the different locations. Some differences occur between the NAVGEM-HA locally analyzed winds compared to the zonal averaged NAVGEM-HA analyzed winds for each latitude of the meteor radar stations. Short-term variations during winter are much more visible in the locally analyzed winds, this is especially true for the meridional wind case.

## 4.2  Mean temperatures

Until now, only NAVGEM-HA wind products have been extensively validated with independent ground-based measurements. To extend this validation to MLT temperatures, we next perform a similar comparison using NAVGEM-HA temperature analyses and a co-located potassium lidar instrument at Kühlungsborn. The composite daily mean lidar temperatures over the period 2003-2012 are shown in Fig. 2 together with the NAVGEM-HA analyzed temperatures between 2009 and 2010. Both data sets show the same seasonal temperature pattern with the lowest temperatures during summer. The mesopause, where the lowest

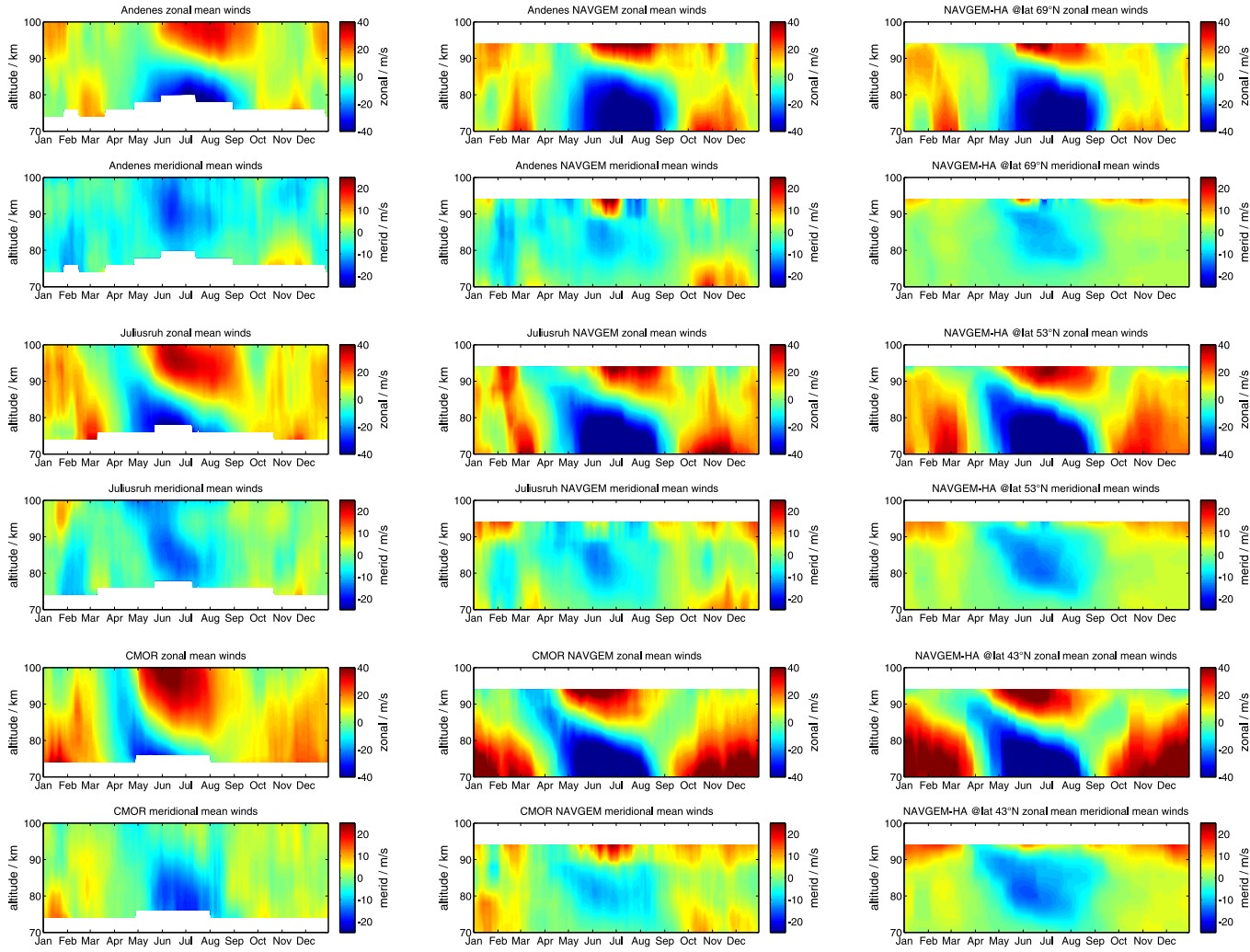

**Figure 1.** Comparison of mean winds above Andenes, Juliusruh and Tavistock (CMOR) using a 30-day running median with periodic boundary condition using the same dates for NAVGEM-HA and the meteor radar observations. The left panel shows the meteor radar observations. The central panel shows the NAVGEM-HA analysis fields for the same locations and periods. The right panel displays the zonal mean zonal and meridional winds for each latitude.

temperatures occur during the year, is estimated from the lidar data and found around 88 km in summer and just above 100 km in winter. For the NAVGEM-HA analyzed temperatures the altitude of the mesopause is in nearly the same altitude range. In general, the temperature values are in very good agreement with each other, although we note that the temperatures observed by lidar near 70 km are larger than the NAVGEM-HA temperature. At the upper edge of the NAVGEM-HA data, there is also a temperature enhancement during summer, which is not seen in the lidar data.

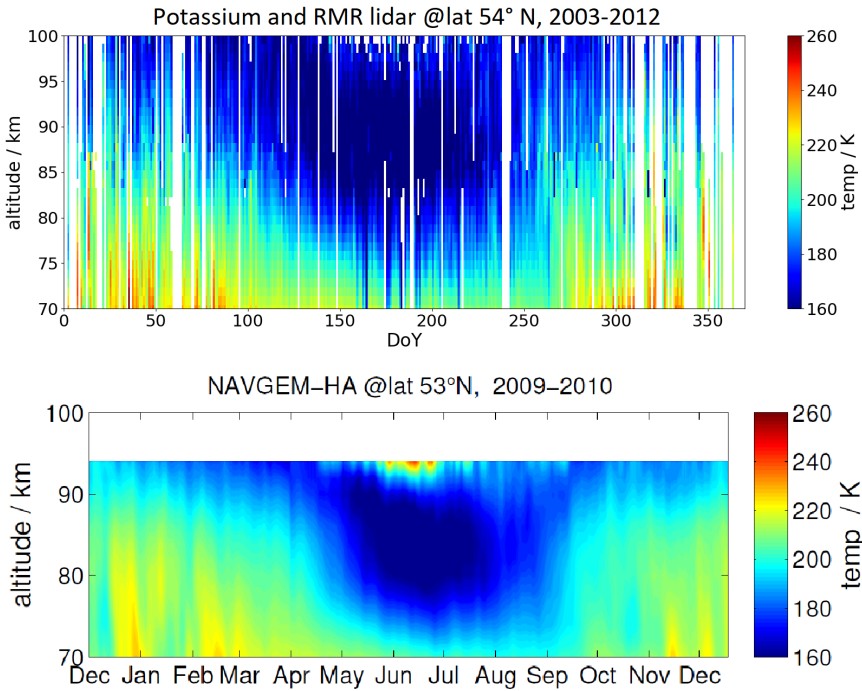

**Figure 2.** Comparison of mean temperatures above Kühlungsborn. The upper panel shows the temperatures derived from the potassium lidar. The lower panel shows the NAVGEM-HA analysis field for the same location.

### 4.3 Semidiurnal tides

In this section, we investigate the seasonal variation of the semidiurnal wind tide based on the calculation with the adaptive spectral filter. This component is the most dominant tidal component at the MLT and the latitudes analyzed herein (Chapman and Lindzen, 1970). The results for the semidiurnal tidal amplitude and phase for the stations at Andenes, Juliusruh and Tavistock are shown in Fig. 3, 4, and 5, respectively. Every data set is compared to the NAVGEM analyzed tidal fields from a local as well as from a global perspective as already done for the mean winds and temperatures.

The observations from all stations indicate a clear winter amplitude maximum. A second maximum is also evident during September. The amplitudes are smallest during November and April. Only Tavistock exhibits a significant semidiurnal tidal amplitude for April compared to the meteor radars at higher latitudes, and we note that the tidal amplitudes above Tavistock are also even stronger during fall than during winter. Compared to the other locations the winter maximum above Tavistock is less pronounced. In general, the amplitudes during winter are strongest for midlatitudes (Juliusruh).

The NAVGEM-HA analyzed amplitudes reveal the same temporal variability over the year as from the observations. Above an altitude of 90 km, the amplitudes from NAVGEM-HA show a significant increase which is not seen in the observations. This was also visible in the temperature data of NAVGEM-HA compared to the lidar data.

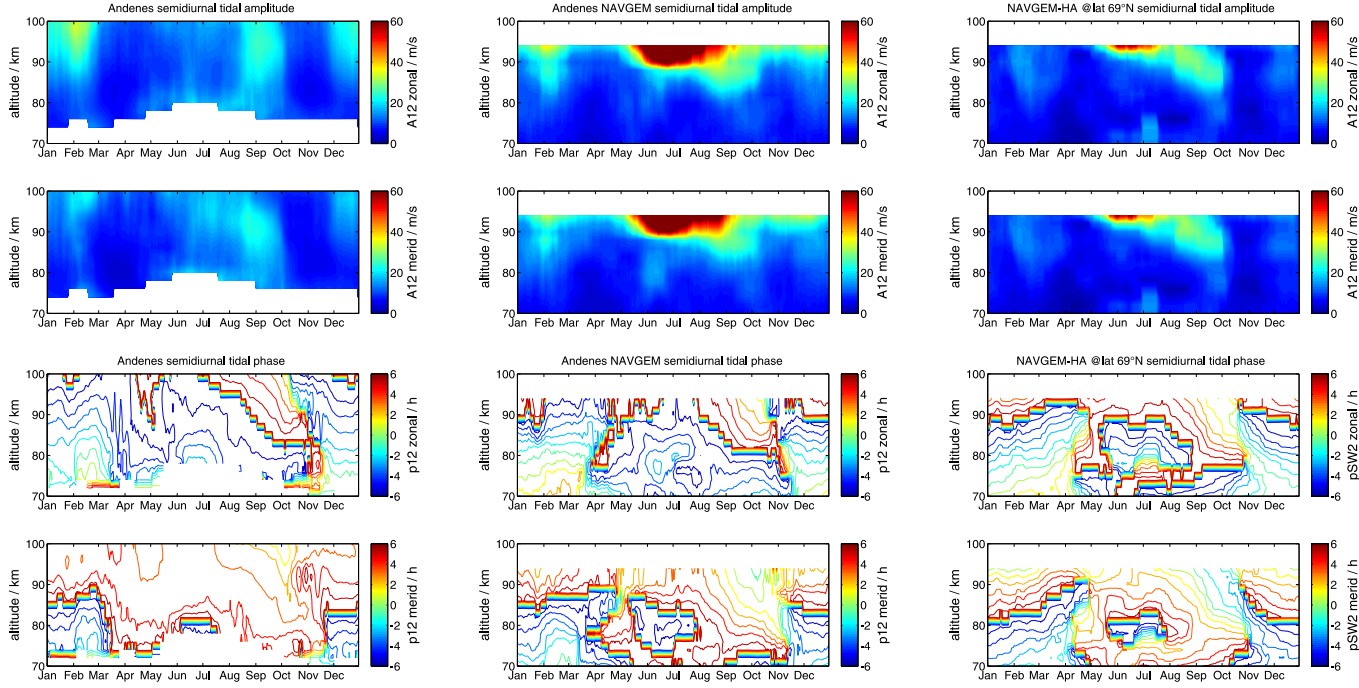

**Figure 3.** Comparison of semidiurnal seasonal zonal and meridional amplitude (upper two rows) and phase (lower two rows) tidal climatology using a 30-day running median with periodic boundary condition. The left panels show the meteor radar observations above Andenes (69° N, 11° E). The central panels show the NAVGEM-HA analysis fields for the same period. The right panels visualize the zonal mean tidal amplitude and phase of the SW2. The label A12 and p12 corresponds to the semidiurnal amplitude and phase using the local diagnostic.

In addition to the amplitudes of the semidiurnal tides, the annual phase behavior of those tides was also calculated using the spectral adaptive filter. In general, for every location, the phase of the semidiurnal tide is drifting and variable over the year. During winter the phases are continuously changing, at the beginning of March, the phase shows a sudden jump, which is evident in every location of the observations and the meteorological analysis. This behavior reverses during October/November, exactly when the atmospheric circulation reverses again from summer to winter conditions. A similar phase progression is visible from the NAVGEM-HA locally analyzed data as well as from the global fields.

### 4.4 Day-to-day variability during a sudden stratospheric warming

Having established that NAVGEM-HA wind and temperature analyses capture many of the salient features in the seasonal variation of both meteor radar wind and lidar temperature observations, we now examine the shorter-term variations in both data sets. Specifically, we examine the day-to-day variability of the mean winds, the semidiurnal tidal amplitudes and phases from the meteor radar winds during the sudden stratospheric warming (SSW) that took place in 2010 and 2013 in comparison to NAVGEM-HA analyzed data from a local perspective as well as from a global view. To do so, we apply the same ASF analysis procedure to both meteor radar and NAVGEM-HA data at a high latitude (Andenes) and midlatitude (Juliusruh) location.

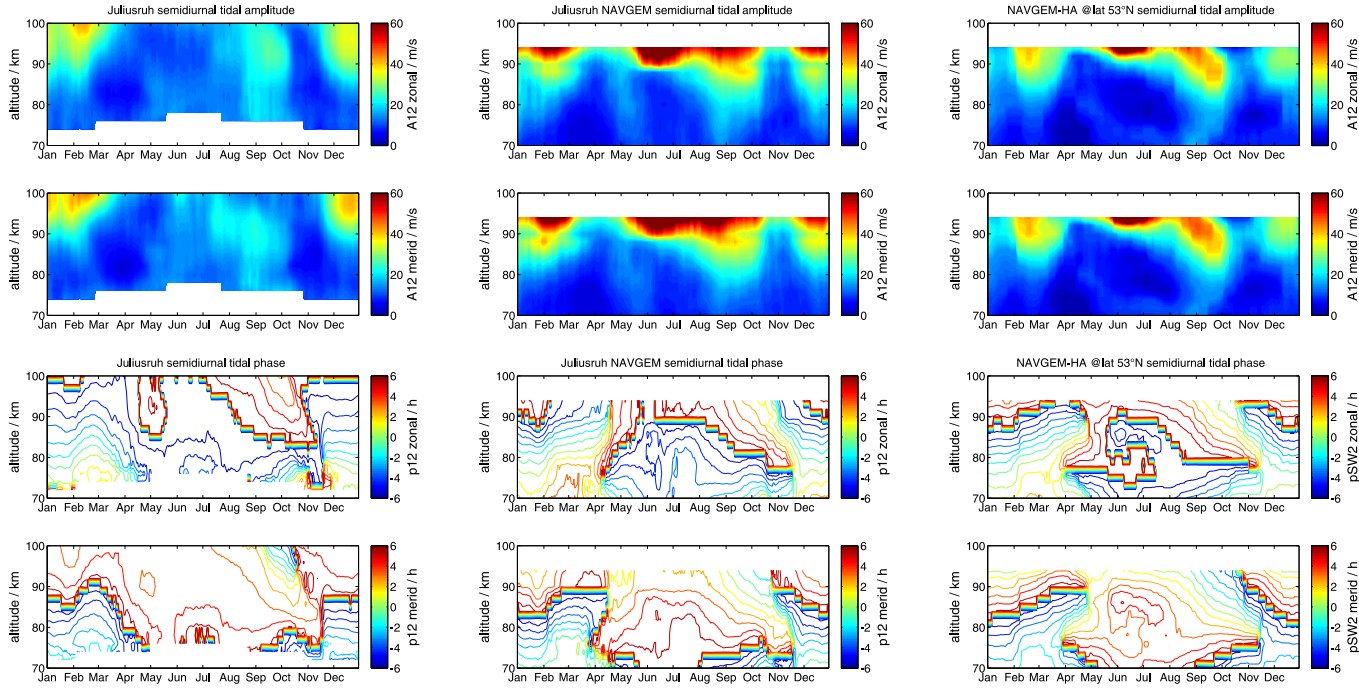

**Figure 4.** Same as Fig. 3, but for Juliusruh.

### 4.4.1 Winter season 2009/2010

During the winter in 2009/2010, a major sudden stratospheric warming occurred at the end of January when the polar vortex was markedly displaced from the pole (Stober et al., 2012) and then separated into two unequally strong lobes (e.g., Dörnbrack et al., 2012; Jones Jr. et al., 2018). Following previous studies involving NAVGEM-HA, we mark the onset of the SSW as

occurring on 27th January (McCormack et al., 2017). Mean winds, the semidiurnal tidal amplitude and phase are shown in Fig. 6 from the meteor radar observations above Andenes as well as for the corresponding locally analyzed NAVGEM-HA data. In Fig. 7 the same results are shown for the station at midlatitudes above Juliusruh. Stronger changes in the winds are visible for Juliusruh than for Andenes. Even the semidiurnal amplitudes are stronger at midlatitudes, which agrees with the stronger seasonal variation of the semidiurnal tidal amplitude above Juliusruh. After the onset of the sudden stratospheric warming, the

semidiurnal tidal amplitudes show an enhancement at the beginning of February, which is visible at both stations and in both wind components.

The semidiurnal tidal phases show a large day-to-day variability during the winter period, which is in general stronger at high latitudes than at midlatitudes. After the central date of the sudden stratospheric warming, the tidal phase shows a sudden increase which lasts only a few days. After these days, the phase shows a recovery where they become more stable again just

as before the sudden stratospheric warming.

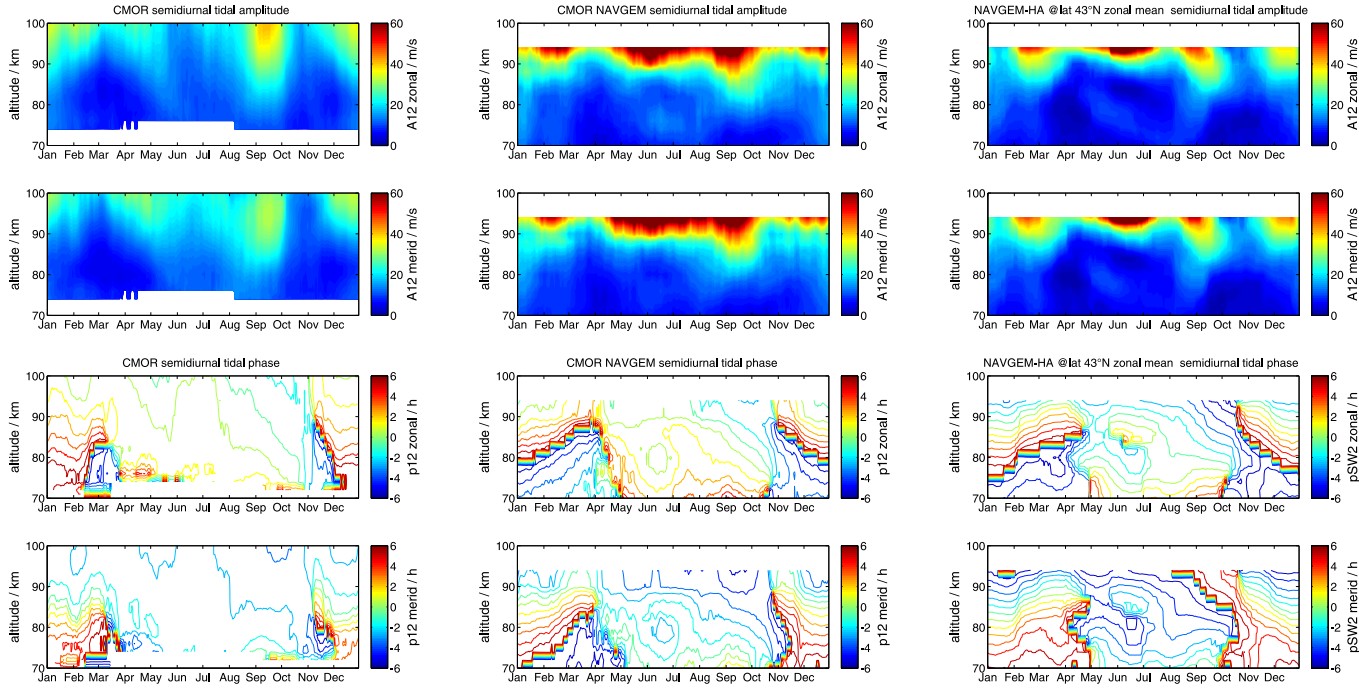

**Figure 5.** Same as Fig. 3, but for CMOR.

The NAVGEM-HA analyzed winds exhibit the same short-term variability during the 2009/2010 winter at both stations for the winds as well as for the semidiurnal tide. Even the phase enhancement after the central date of the SSW is remarkably well reflected by the NAVGEM-HA data. Some differences occur above an altitude of 85 km, where NAVGEM-HA data reveals larger magnitudes in the winds as well as larger amplitudes for the semidiurnal tide as previously seen. Fig. 8 shows global NAVGEM-HA results for both Andenes and Juliusruh station locations. The global analyzed NAVGEM-HA data indicates much less variability during the winter compared to the locally analyzed data. But the central date of the SSW is more easily identified in the winds than it was the case for the locally analyzed winds. However, the main features for the semidiurnal tide stay the same. The amplitudes show an increase after the central date of the SSW and the phases reveal a change for a few days at both locations. In contrast to the locally analyzed data, the phases from the global NAVGEM-HA fields slowly change during the winter. But in general, the agreement with the MR observations is still good.

### 4.4.2 Winter season 2012/2013

The winter season in 2012/2013 was also characterized by a major sudden stratospheric warming. In this case, the onset of the SSW occurred on 7th of January using again the definition presented in McCormack et al. (2017). During the SSW the vortex was split into two lobes (Coy and Pawson, 2015). Again, mean winds, the semidiurnal tidal amplitudes and phases are shown

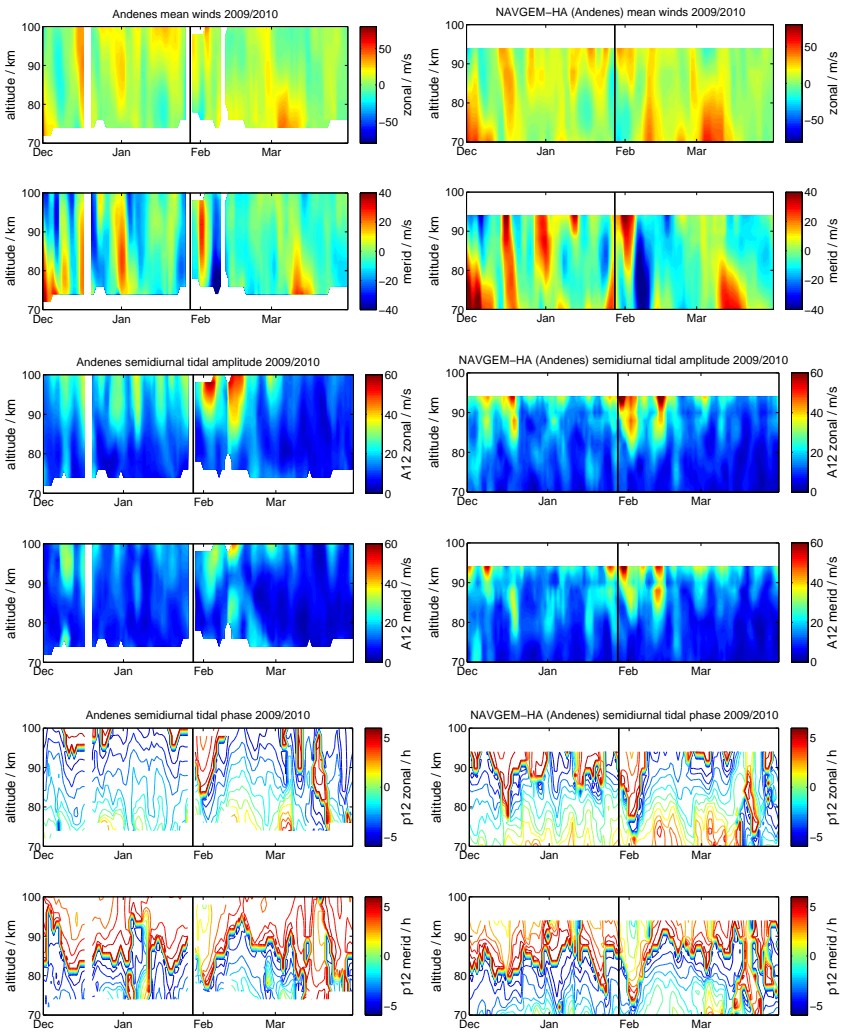

**Figure 6.** Comparison of meteor radar observations and NAVGEM-HA above Andenes during the winter 2009/10 for daily mean zonal and meridional winds (upper two panels), semidiurnal tidal zonal and meridional amplitude (middle panels) and semidiurnal tidal phases (lower two panels). The label A12 and p12 corresponds to the semidiurnal amplitude and phase using the local diagnostic.

in Fig. 9 and Fig. 10 for high latitudes and midlatitudes, respectively.

In this winter season, the mean zonal winds at high latitudes are stronger, especially after the SSW, than at midlatitudes, which is opposite to that seen in the winter season 2009/2010. The mean meridional winds are similar in strength for both

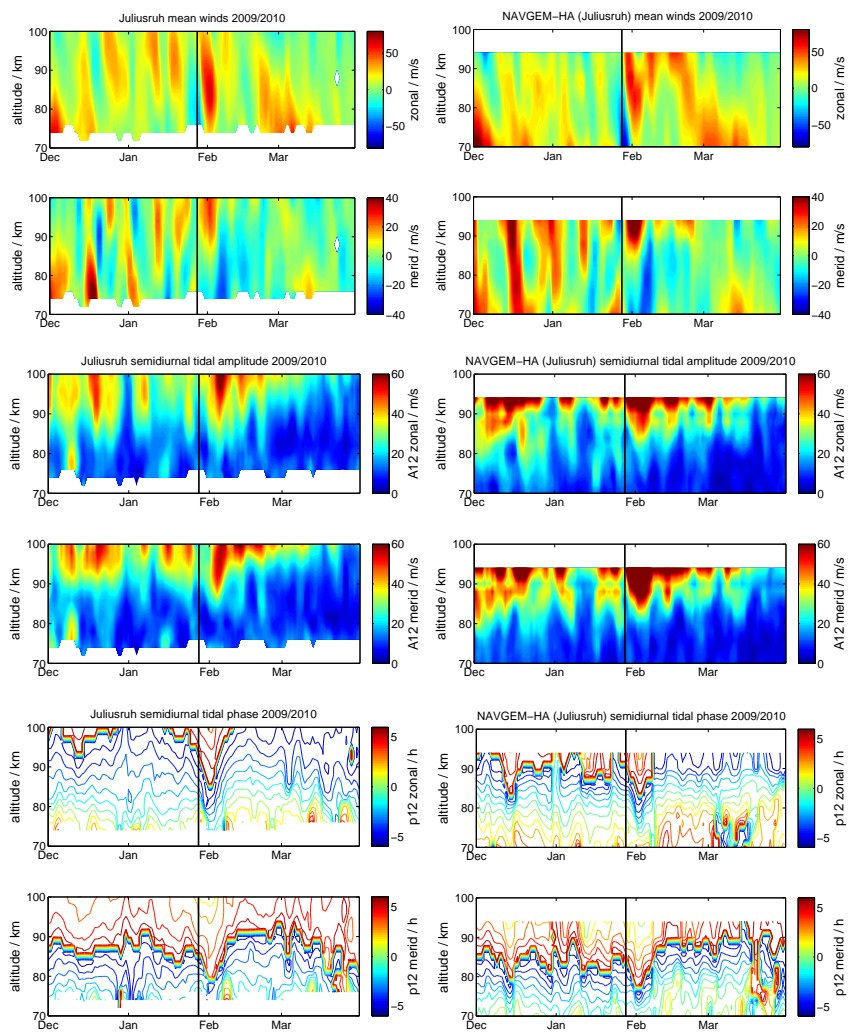

**Figure 7.** The same as Fig. 6, but for Juliusruh.

stations. Nevertheless, the semidiurnal tide shows again stronger amplitudes at the midlatitude station than at high latitudes. At Andenes, we see a distinct increase of the amplitudes after the SSW, which was already seen in the winter season 2009/2010, while in general at Juliusruh a larger tidal activity is visible. Here, before and during the central date of the SSW the tidal amplitudes decrease in the first place due to the strong changing winds. Afterward, the semidiurnal tidal amplitudes increase
5 again stronger than during the whole winter.

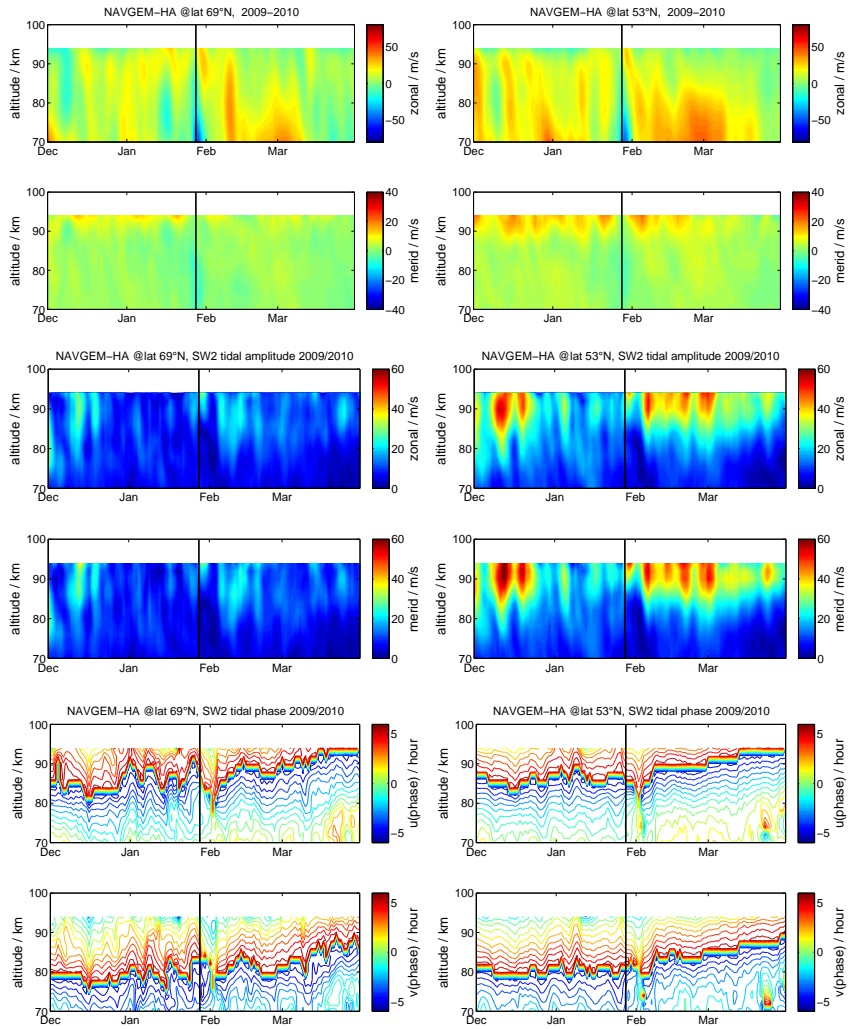

**Figure 8.** Comparison of global NAVGEM-HA above Andenes and Juliusruh during the winter 2009/10 for daily zonal mean zonal and meridional winds (upper two panels), zonal mean semidiurnal tidal zonal and meridional amplitude (middle panels) and zonal mean semidiurnal tidal phases (lower two panels).

The semidiurnal tidal phases show again a large variability during the whole time. A few days after the central date of the sudden stratospheric warming a sudden increase of the phase is visible in the same way as for the winter 2009/2010. The locally analyzed NAVGEM-HA data reveal structures during this winter period similar to those from the observations. Evident from every data set, the mean winds, as well as the amplitudes, are slightly overestimated in NAVGEM-HA. The NAVGEM-HA

analyzed tidal phases exhibit also a sudden change after the SSW, but not as strong as from the observations, but this might be due to a general more disturbed winter period compared to the year 2009/2010. The globally analyzed data from NAVGEM-HA are shown in Fig. 11. As was the case for the 2009/2010 winter, in this winter season the winds from a global perspective are much stronger and uniformly distributed over the winter months, except for the wind reversal during the SSW, which is visible at the beginning of January.

## 5 Discussion

### 5.1 NAVGEM-HA and MR mean wind and temperature climatology

The comparison of the NAVGEM-HA mean winds and the meteor radar climatologies at Andenes, Juliusruh and CMOR is remarkable up to an altitude of 94 km. The assimilation of satellite-based middle atmospheric temperature and constituent observations enables NAVGEM-HA to capture the main features of the seasonal wind climatologies such as the weak eastward winds during the winter, the asymmetry of the seasonal pattern between the spring and autumn wind reversals as well as the gradual descent of the summer wind reversal between the mesospheric westward winds and the higher altitude thermospheric eastward jet. Our analysis shows that the initial good agreement reported during the winter months in McCormack et al. (2017) extends to seasonal time scales, and providing further cross-validation of the NAVGEM-HA winds with globally distributed and available meteor radar wind observations.

The MLT mean wind climatology is still afflicted with a high degree of uncertainty when comparing different GCMs, although nudged to the same reanalysis data sets. Pedatella et al. (2014) compared several GCMs and showed that not even the sign of the mean wind seems to agree between the models at the MLT. Further, the seasonal morphology at mid- and high-latitudes was not well-reproduced by some models compared to the climatologies published from meteor radars (Portnyagin et al., 2004; Wilhelm et al., 2019). In particular, the seasonal asymmetry of the zonal wind circulation from the winter to the summer conditions and back to the winter regime seems to be problematic for the GCMs. Comparing the seasonal morphology of the zonal and meridional winds between NAVGEM-HA and other comprehensive GCMs, such as WACCM or SD-WACCM (Smith, 2012; Chandran and Collins, 2014), and the meteor radar and lidar data indicate a much better agreement for the meteorological analysis for altitudes beyond 80 km. Similar results have been found by comparing meteor radar winds to free running mechanistic GCM (Pokhotelov et al., 2018). Previous studies comparing eCMAM with ground-based meteor radar observations at low latitudes and on seasonal time-scales revealed a similar good agreement for mean winds and diurnal tidal amplitude and phases (Du et al., 2007; Ward et al., 2010). The good agreement between NAVGEM-HA and ground-based wind observations shown in Section 4 indicate that global data assimilation products in the MLT can provide a valuable benchmark for evaluating the performance of "whole-atmosphere" GCMs extending into the thermosphere. These products could improve understanding of the large discrepancies among different models noted above by offering insight regarding where these models deviate most from a validated high-altitude meteorological analyses.

The general thermal structure and seasonal climatology are also well-reproduced in NAVGEM-HA for the lidar observations presented in Fig. 2 at the mid-latitude station of Kuehlungsborn. The meteorological analyses captures the seasonal course of

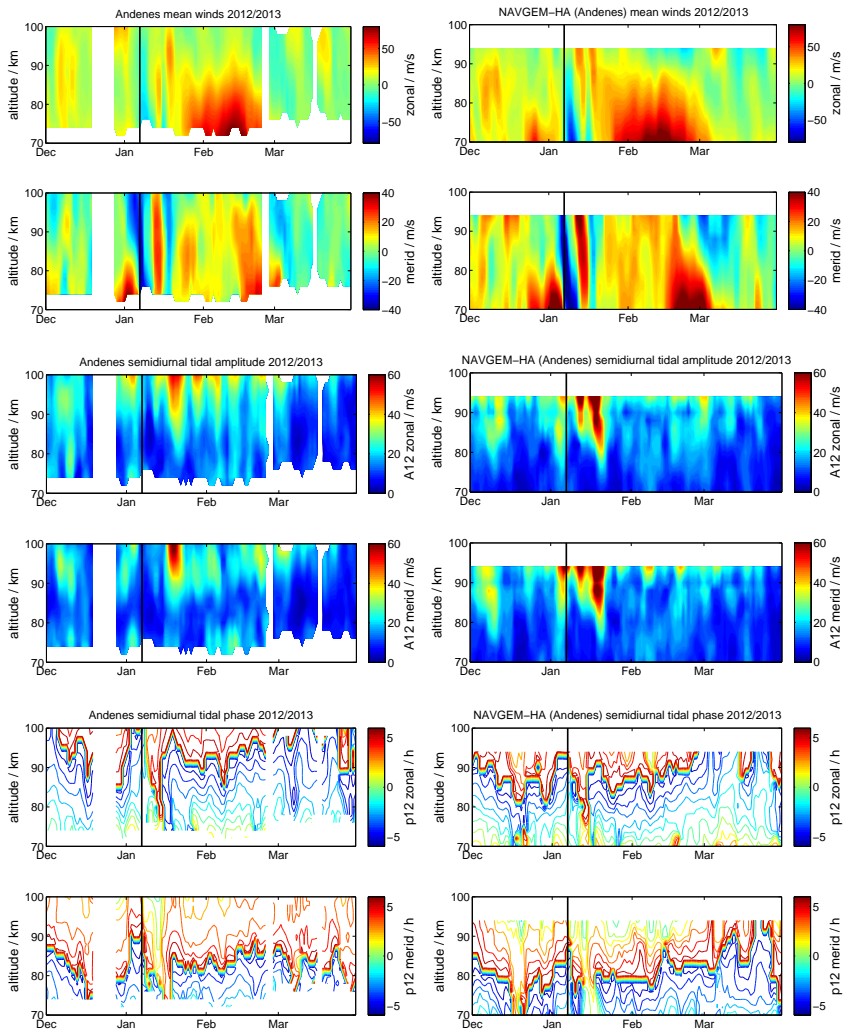

**Figure 9.** Comparison of meteor radar observations and NAVGEM-HA above Andenes during the winter 2012/13 for daily mean zonal and meridional winds (upper two panels), semidiurnal tidal zonal and meridional amplitude (middle panels) and semidiurnal tidal phases (lower two panels).The label A12 and p12 corresponds to the semidiurnal amplitude and phase using the local diagnostic.

the altitude variation of the mesopause. Further, we identified a small offset between the lidar and the NAVGEM-HA temperatures. The analysis data has a tendency towards slightly warmer temperatures compared to the resonance lidar. These slightly higher temperatures in NAVGEM-HA may also explain the higher wind magnitudes relative to the MR observations.

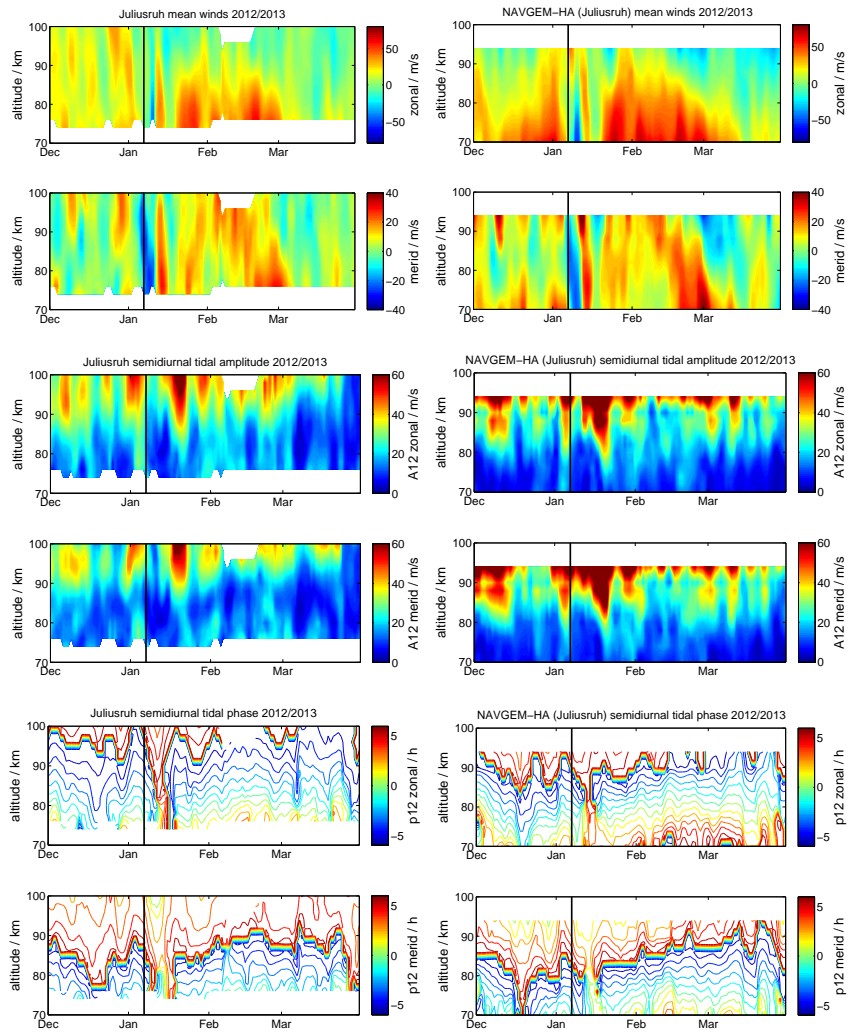

**Figure 10.** The same as Fig. 9, but for Juliusruh.

## 5.2 NAVGEM-HA and MR mean wind semidiurnal tidal comparison

At MLT heights, tidal amplitudes grow large and contribute significantly to the daily variability of the zonal and meridional winds. At mid- and polar latitudes, the semidiurnal tide is the most prominent tidal wave in the MLT that can be observed throughout the course of the year (Portnyagin et al., 2004; Pokhotelov et al., 2018; Wilhelm et al., 2019).

5  In principle, local observations (single measurements) can not distinguish between migrating and non-migrating tidal com-

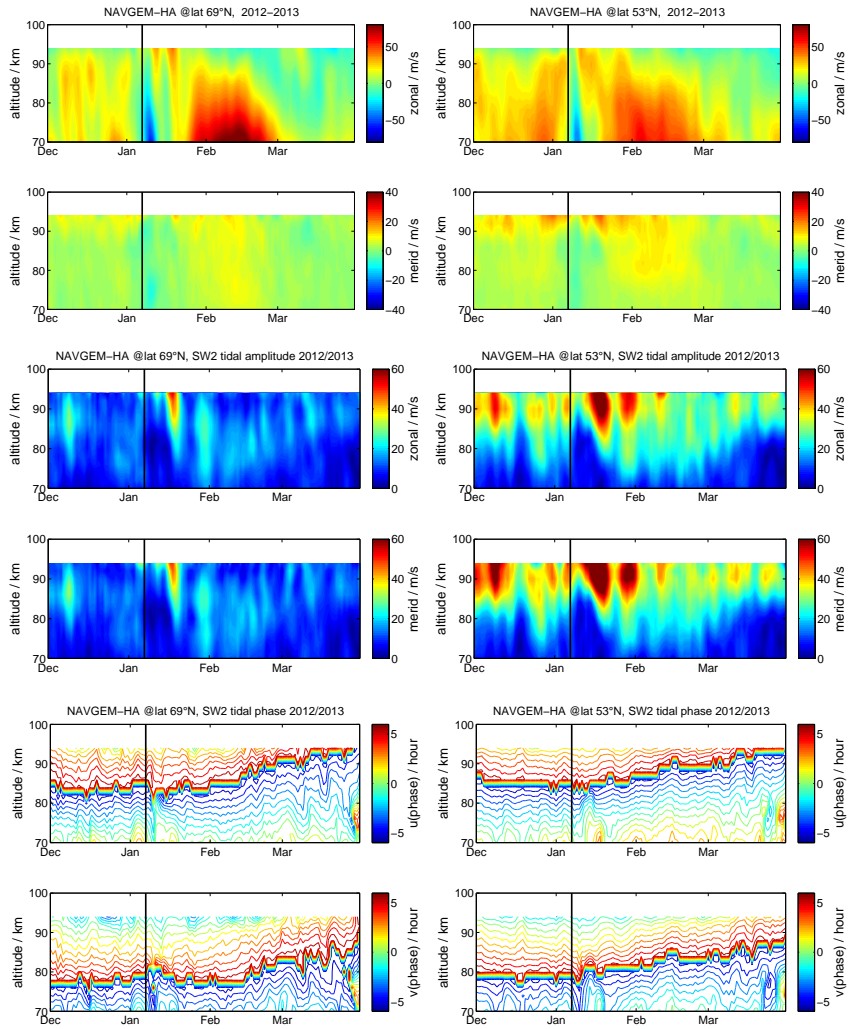

**Figure 11.** Comparison of global NAVGEM-HA above Juliusruh (left column) and Andenes (right column) during the winter 2012/13 for daily zonal mean zonal and meridional winds (upper two panels), zonal mean semidiurnal tidal zonal and meridional amplitude (middle panels) and zonal mean semidiurnal tidal phases (lower two panels).

ponents and only observe a total tide. Earlier studies (e.g., Portnyagin et al., 2004) have investigated the global nature of the diurnal and semidiurnal tide at polar latitudes using a chain of radars at approximately 70° N. They found very good agreement between monthly tidal amplitudes and phases for all stations along the latitudinal circle. Recently, there have been some attempts to separate migrating and non-migrating tides using globally distributed chains of meteor radars (He et al., 2018) as-

suming theoretical tidal wave fields consisting of migrating and non-migrating components. However, due to the small number of meteor radars at the latitudinal circles, the analysis still contains a high degree of ambiguity.

Combining the benefits of high resolution local measurements with global meteorological analysis data solves this problem. The comparison of the semidiurnal tidal climatology reveals that NAVGEM-HA reproduces the seasonal morphology of the tidal amplitudes for both wind components up to an altitude of 90 km applying the ASF tidal diagnostic. The local ASF diagnostic shows remarkable agreement between the global tidal analysis of the migrating SW2 tide in magnitude and phase. The non-migrating semidiurnal components show only very small and often negligible amplitudes. The agreement of the phase of the SW2 tide between the global and local measurements seem to be better at lower latitudes of CMOR and Juliusruh compared to Andenes.

Ward et al. (2010) performed a similar validation of the tidal amplitude and phase behavior using the extended Canadian Middle Atmosphere Model (eCMAM) at low latitudes. However, they used much longer windows of 60-days to compute average amplitudes and phases. They also found the seasonal change of the tidal phase and remarkable good agreement between the ground-based lidar and radar data and the model. The comparison of NAVGEM-HA and the meteor radar indicates that tidal phase are variable on the seasonal scale showing already significant shifts and drifts within a week. Previously, for local observations this phase variability was assumed to be the result of a superposition of migrating and non-migrating tidal modes. However, comparing the global tidal fit obtained from NAVGEM-HA of the SW2 tide reflects this behavior as well. These continuous phase changes have severe implications for the analysis of tides at mid- and high latitudes from satellites, which usually requires to average over several weeks to cover all local times.

### 5.3 NAVGEM-HA and MR winds and tidal day-to-day variability and lunar tides during SSW events

Besides comparing mean winds, we also investigated the day-to-day variability of the semidiurnal tide during two winter seasons with a major SSW event at the mid-latitude location Juliusruh and polar latitudes above Andenes. In 2010 there was a vortex displacement event (e.g., Stober et al., 2012; Matthias et al., 2013), which was already validated by a cross comparison of the mean winds and waves in McCormack et al. (2017) using several worldwide-distributed meteor radars. The second SSW event occurred during winter 2012/13 and evolved as a vortex splitting event (e.g., Xu and San Liang, 2017).

Daily mean winds and tidal amplitudes were diagnosed by the ASF. The meteorological analysis of NAVGEM-HA reproduces the general day-to-day variability of winds and even shows a high level of agreement for individual planetary waves passing over the stations. In particular, the timing of the SSW event itself with the zonal wind reversal and the formation of an elevated stratopause is well-captured. Similar to the zonal and meridional wind climatologies, the meteorological analysis tends to show higher magnitudes of the wind speeds. Previous comparison of wind observations to model data, such as ECMWF or MERRA2, were limited to a maximum altitude of approximately 70-75 km and below (Rüfenacht et al., 2018) and, thus, we omit here any further detailed discussion.

Another very important aspect of this study is the phase variability on a day-to-day basis. The ASF provides information on the phase stability of tides with basically the same resolution as the original measurement time series. Very often tidal phases are assumed to be stable over long periods of up to several months in the analysis. However, for instance, the TIMED satellite

requires 60-days to cover all local times due to its orbit geometry (Zhang et al., 2006; Oberheide et al., 2011). Our results indicate that during an SSW the phase of the SW2 tide is significantly altered on a global scale as well as on a regional or local scale as the dynamics of the middle atmosphere change (e.g., Manney et al., 2009; Matthias et al., 2012). Fuller-Rowell et al. (2016) discussed three possible mechanisms to understand these changes of the tide; Fuller-Rowell et al. (2010); Jin et al.

(2012) attributed the change of the migrating tidal phase to changes of the mean winds in the middle atmosphere, whereas Pedatella and Forbes (2010) suggested non-migrating tides as a source of the SW2 phase variability. Other studies favor an amplification of the lunar tide during an SSW (Fejer et al., 2010; Forbes and Zhang, 2012). We discuss these three aspects using the results obtained from the ASF decomposition of the local and global measurements and meteorological analysis data with a particular emphasis on the suggested lunar tide amplification. Thus, we are introducing a holographic analysis and lunar

orbit parameters as proxy of the lunar forcing. Furthermore, we are determining the zonal and vertical wavenumbers as well as the period of the semidiurnal tide and their temporal evolution to separate a potential lunar forcing from the solar driven semidiurnal tide.

At first, we investigate the non-migrating tidal components (see appendix B1,B2,B3,B4) derived from NAVGEM-HA winds. It appears that only the SW1 and SW3 tides show a response to the SSW event depending on the latitude and how the SSW

evolved. This is consistent with previous studies (Du et al., 2007; Liu et al., 2010). The SE1, SE2, SE3 and S0 semidiurnal tides show much smaller amplitudes and are negligible compared to the SW2 tide, in particular, at polar latitudes. Another interesting aspect when comparing the migrating and non-migrating tides from NAVGEM-HA is the winter seasonal phase behavior of the SW2 tide. The phase of the tide drifts by several hours between December to March, which correlates to the mean wind morphology. Apparently, the change of the phase of the semidiurnal tide is not explained by a superposition of

migrating and non-migrating tides. However, this needs to be examined in more detail and is beyond the scope of this paper.

Secondly, we are discussing in detail the suggested lunar tide amplification after an SSW introducing a holographic diagnostic and the lunar orbit elevation as proxy of the lunar forcing. Fejer et al. (2010) investigated vertical plasma drifts above Jicamarca and found a drift in local time of the semidiurnal oscillation during SSWs, which was attributed to the lunar tide assuming that all other tidal waves remained stationary and monochromatic. Later, Forbes and Zhang (2012) proposed that the lunar tide

enhancement is a result of the Pekeris resonance effect that is shifted towards the lunar tide period M2 of 12.42 h due to changes in the mean zonal winds and vertical temperature structure caused by the SSW. They tested the proposed physical mechanism on satellite observations from SABER, CHAMP and GRACE and the steady state Global-Scale-Wave-Model (GSWM) for a case study and the SSW in 2009. To separate the lunar tide from the semidiurnal SW2 tide, they used a window of 24-days to ensure sufficient frequency resolution and assumed monochromatic and stationary tidal waves within the window. Later,

Zhang and Forbes (2014) claimed that the lunar tides seem to enhance during nearly every SSW event arguing that the Pekeris resonance has a rather broad peak and, thus, the resonance conditions are satisfied for all SSW events, although, Forbes and Zhang (2012) pointed out that a very specific thermal and dynamic structure is required to satisfy the resonance condition.

In the following, we investigate the phase variability of the semidiurnal tide introducing a holographic analysis for the SSW 2012/13 and discuss a potential connection to the Pekeris resonance (Zhang and Forbes, 2014). Similar to other holographic

analysis, we use the phase differences between a coherent reference wave and the observed wave field, that propagated through

the atmosphere, to infer small deviations in frequency that are not resolvable by standard Fourier techniques. The day-to-day variability, obtained from the ASF, indicates that the tidal phase are not stable with time and show significant interday variability, which appears to be related to changes in the zonal wind in the middle atmosphere driven by the polar vortex and planetary waves. Considering that a time dependent phase corresponds to a frequency shift, it is possible to convert

this temporal phase variability into a period change and, hence, to estimate the spectral line shape of the tide or to derive a holographic representation of the temporal evolution on a day to day basis.

The hologram is derived considering that the tide can be represented by a cosine wave with amplitude $A$ (e.g., semidiurnal tide), a mean frequency $w$ and a time dependent phase $\phi(t)$;

$$A(t) = A\cos(wt + \phi(t)) \ . \tag{3}$$

Although the true functional form of the time dependent phase might be unknown, we can express this function as a Taylor series at a certain point in time $t$;

$$\phi(t) = \phi_0 + \frac{d\phi}{dt} \cdot t + .... \tag{4}$$

Truncating the Taylor series at the first order and inserting them in eq. 3 leads to;

$$A(t) = A\cos(wt + \phi_0 + \frac{d\phi}{dt} \cdot t) \ . \tag{5}$$

Rearranging the terms according to their time dependence leads to;

$$A(t) = A\cos((w + \frac{d\phi}{dt}) \cdot t + \phi_0) \ . \tag{6}$$

It is now straight forward to numerically obtain the time dependent phase change $d\phi/dt$ using a central differences approach in the complex domain.

In Fig. 12, we show a holographic reconstruction based on the ASF decomposition of the semidiurnal tide. This technique only assumes monochromaticity within the adopted window length (less than a day for the semidiurnal tide) and, thus, captures non-stationary processes on an interday basis. The hologram shows that during the SSW event in 2012/13 the phase behavior of the semidiurnal tide itself is shifted to the period range that is expected for the lunar tide M2 (solid white line) and N2 (dashed white line). Further, we overlayed the lunar orbit as elevation angle for the geographic location of Juliusruh to search

for a potential connection of the semidiurnal phase variability and the moon position on the sky. The hologram for the global diagnostic is shown in Fig. 13. The main differences in the holograms between the local MR observations and the global tidal fields from NAVGEM-HA are attributed to the decomposition of the global fields into migrating and non-migrating tides. As shown in the appendix B1,B2,B3,B4, there is an excitation of the non-migrating tides SW1 and SW3, which leads to the differences in the holographic reconstruction. The local diagnostic shows the superposition of all tidal components. However,

the global diagnostic also indicates the frequency shifts of the SW2 tide to periods that can match the predicted Pekeris lunar tide resonances. However, as indicated by the white lines, these phase shifts happen frequently during a winter season and are

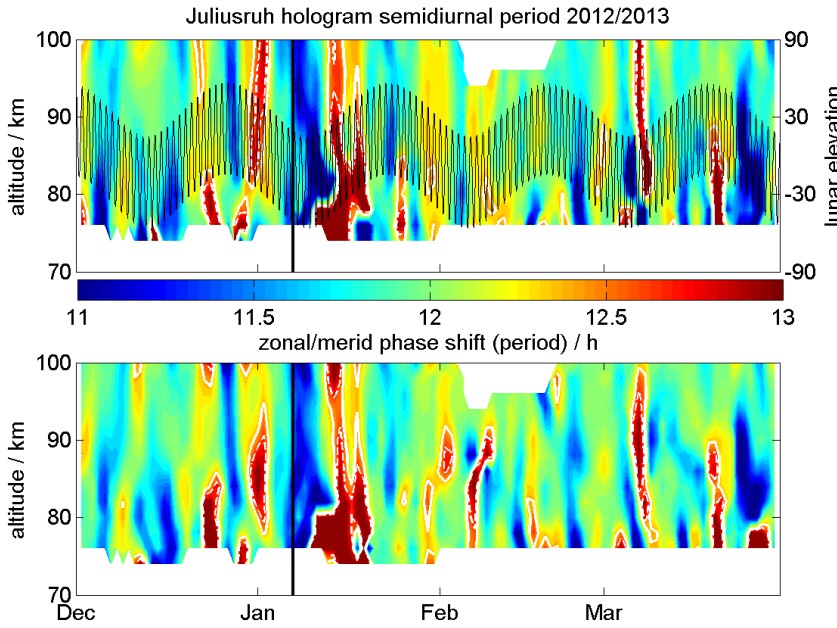

**Figure 12.** Holographic reconstruction of the semidiurnal tidal phase variability. The hologram shows the periods using a time variable phase, which is equivalent to a frequency shift or change in period. The white contour lines indicate the lunar tide M2 (12.42 h) (solid line) and N2 (12.66 h) (dashed line). The lunar orbit as elevation angle (-90° to 90°) for Juliusruh is plotted as black solid line.

neither correlated to the lunar orbit nor accompanied by a tide enhancement. The effect of the SSW is visible in both holograms up to 10 days after the onset of the SSW, which is also the time delay corresponding to the amplification of the semidiurnal tide after the SSW. Moreover, Forbes and Zhang (2012) reported a delay of 5-7 days between the occurrence of the lunar tide amplification and the central day of the SSW event. This delay of approx. 5-7 days is consistent with the holographic analysis,

which also shows that the frequency/period shift towards the lunar tide frequency/period (M2 and N2) occurs after the SSW event (central day), at the beginning of the formation of an elevated stratopause or when the polar vortex begins to restore, however, well before the semidiurnal tide enhancement. This time span also corresponds to the response time of the semidiurnal tide to a transient forcing, which was estimated to be between 6-10 days (Vial et al., 1991) for comparisons to steady state models.

Many recent studies have investigated lunar tides with window lengths that are long enough to ensure an unambiguous frequency resolution to separate the lunar tide from the semidiurnal tide, which requires at least 21-days or more (e.g., Forbes and Zhang, 2012; Chau et al., 2015; Conte et al., 2017; He et al., 2018; Siddiqui et al., 2018, and reference therein). The ASF analysis indicates that there is a considerable interday tidal variability in amplitude and phase, which poses a challenge to the signal processing. Such intermittent behavior suggests that long windows (longer than even a day) might lead to spurious

results, and do not allow a separation of the different waves from each other. The zonal wind reversal and accompanied cooling

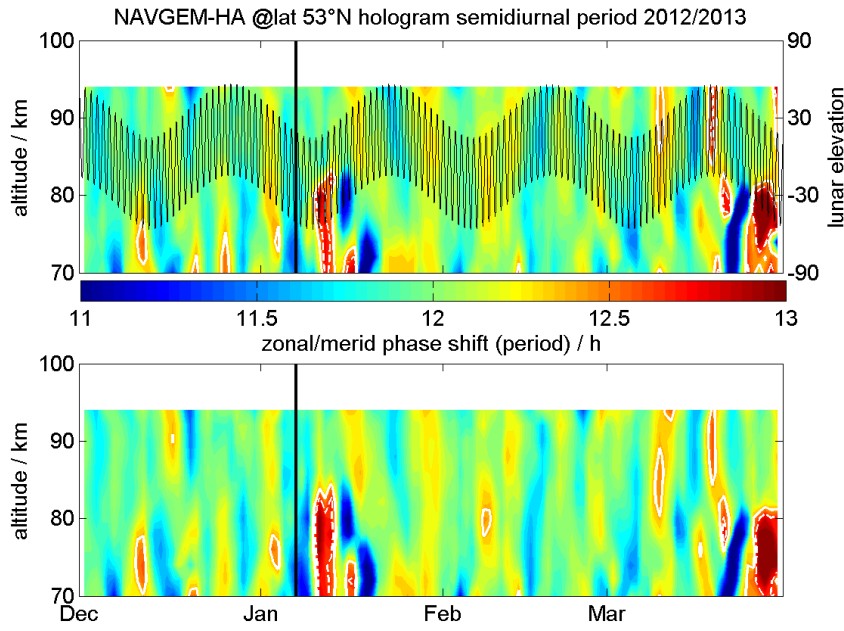

**Figure 13.** The same as Fig. 12, but for the global tidal analysis of the SW2

.

at the MLT during a SSW last only for a few days (much shorter than the typical window length used for the lunar tides) and cause significant changes in the zonal mean wind at mid- and polar latitudes altering the propagation conditions for the tides. As a consequence, such a long window would not allow one to capture SSW effects, which themselves cause changes in the semidiurnal tide. Thus, if one does not notice that an SSW occurred, one cannot know whether the 12.42 h tide is lunar or a

semidiurnal tide that was altered by the SSW.

These shortcomings are also mentioned and discussed in (Forbes and Zhang, 2012; Zhang and Forbes, 2014). They fitted the lunar tide, for instance, on the residuals of SABER measurements after removing the semidiurnal tide by a 12-day running mean, which still is too long given the huge phase variability of the tide. The caveats of the steady state model GSWM are also discussed in Forbes and Zhang (2012). In particular, the steady state assumption seems to be not fully met during an SSW

recalling the results from Vial et al. (1991) as the whole event lasts only 3-4 days at the MLT.

The next aspect we did investigate is a potential correlation between the phase shifts from the holographic analysis and the lunar orbit. Thus, the lunar azimuth, elevation and lunar distance was evaluated, whether there exists a connection with the SSW events. Holographic analysis of the S2 phase shifts shows frequently periods close to the the M2 tide, indicated by the white contour lines, but no obvious correlation to the lunar position or the other orbital parameters were found. This behavior

is also reproduced for the global hologram.

Finally, we examined the properties of the semidiurnal tide with respect to the frequency and vertical wavelength at the MLT

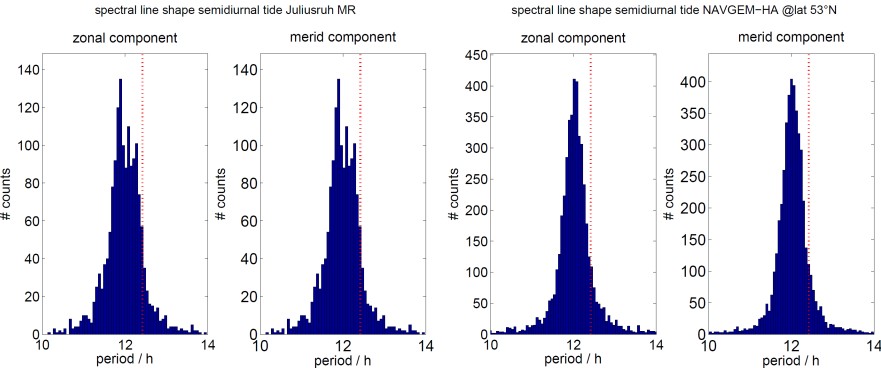

**Figure 14.** Histograms for the zonal and meridional frequency shift due to a temporal variable phase derived from the holograms from December 2012 until March 2013. The left two panels show the local meteor radar observations at Juliusruh and the right two panels the global diagnostic inferred from NAVGEM-HA.

before and during the SSW as well as during the amplitude enhancement after the SSW. Mainly to underline that we can derive all wave properties of the SW2 tide combinin glocal and global diagnostics, which are the zonal wave number, the vertical wavenumber and the frequency on a interday basis. The vertical wavelength of the semidiurnal tide was about 50-60 km before the SSW at altitudes between 74-100 km (see appendix C1). The same vertical wavelength was observed during

the tide enhancement. Whereas during the SSW event the vertical wavelength suddenly increased to 150-600 km at Juliusruh and then decreased just as suddenly after the wind reversal, but before the tide enhancement. The global zonal mean diagnostic exhibit similar vertical wavelengths before and after the SSW, but the sudden response in vertical wavelength due to the SSW was less pronounced. However, the hologram also shows that the mean period of the S2/SW2 tide before the SSW and during the tide enhancement is centered around 12.0 h. This does not indicate a lunar tide enhancement due to the Pekeris resonance

effect, which would require a 12.42 h period. Only during the SSW event itself, there is an increase in the vertical wavelength and the shift of the period towards the M2 period visible pointing to a lunar tide signature that could be interpreted as Pekeris resonance. However, this needs to be investigated with comprehensive models to account for the complex dynamics associated with a SSW. Previous analysis of the lunar tide facilitating multi-year observations from meteor radars by Sandford et al. (2006) showed that the signal is much weaker compared to the total S2 tide. Their spectral analysis also confirms that tides show some

spectral broadening. Such a line broadening is also found in our holographic analysis.

     Fig. 14 shows the histograms of the frequency distributions obtained from the holograms. The left two panels are computed from the meteor radar observations at Juliusruh (zonal and meridional) and the right two panels from the global diagnostic using NAVGEM-HA at the same latitude (zonal and meridional, respectively). The spectral line shape seems to agree from their general morphology, in particular, the line width. The vertical dashed line denotes the period of the lunar tide M2, which lies

in the natural line width of the SW2 tide. However, the peak of the spectral line obtained from the meteor radar observations at Juliusruh shows two side peaks that can be associated to the vortex splitting event and are related to the planetary wave activity

during the winter 2012/13. Due to instrumental effects the number of measurements is not equally distributed over the winter season leading to an apparent double peak structure (this was validated looking at other meteor radar data that are not used in this study). The same plot obtained from NAVGEM-HA at the Juliusruh location shows a fully symmetric spectral line shape similar to the global diagnostic. During the vortex displacement event in the winter season 2009/10 the spectral line at Juliusruh

is entirely symmetric similar to the global diagnostic for both cases. The global diagnostic is not prone to this type of effect as all longitudes are included in the analysis and, hence, these particularities average out. In the case of lunar tide amplification the global diagnostic should reveal a shoulder at the M2 period or asymmetry around the dashed vertical line, which seems to be not present.

Finally, and similar to previous studies (Fuller-Rowell et al., 2010, 2016), we attribute the day-to-day variability of the semid-

10 iurnal tidal amplitudes and phases to changes of the zonal winds in the middle atmosphere altering the vertical propagation conditions. Although atmospheric tides are global scale waves, their vertical propagation depends on the regional meteorological situation. As a consequence, the observed period or phase at the MLT can be altered. Due to the long horizontal wavelength of the semidiurnal tide, a change in the wind pattern in the middle atmosphere manifests as changes in phase for a single station measurement accompanied by a change of the vertical wavelength of the tide. The holographic reconstruction shows that the

15 day-to-day variability of phase is equivalent to a Doppler shifting of the intrinsic tidal frequency, which causes the line broadening at the MLT. Furthermore, an SSW is also associated by a large exchange of air masses between mid- and polar latitudes, which leads to a significant enhancement of the ozone volume mixing ratio inside the polar cap (Schranz et al., 2019), and, thus, provides a potential source to increase the tidal forcing of the semidiurnal tide. Further, this strong meridional coupling also provides a sufficient strong response to explain the low latitude response to the SSW. This aspect needs to be researched in

more detail, but provides a a reasonable approach to explain the semidiurnal tide enhancement after the SSW and the observed low latitude responses (Fejer et al., 2010).

## 6   Conclusions

In this study, we cross-validate NAVGEM-HA meteorological analyses with ground-based meteor radar and lidar observations

at mid- and high latitudes. For the validation, we performed a detailed analysis of mean winds and temperatures and atmospheric tides using a recently developed tool called adaptive spectral filter (ASF), which is designed to capture the intermittent tidal behavior and provide vector information for mean winds and tides for climatologies. We present a comparison of mean winds, temperatures and the semidiurnal wind tide and its phase behavior and a detailed discussion of the day-to-day variability of the semidiurnal tide during two SSW events in 2009/10 and 2012/13 combining global and local diagnostics. We

discussed our results in the context of previous studies, in particular, on the lunar tide amplification during SSWs and have outlined potential issues due to the day-to-day semidiurnal tidal variability. The agreement between MR/lidar climatologies and NAVGEM-HA analysis data is remarkably good compared to the seasonal wind and temperature pattern of comprehensive models. NAVGEM-HA tends to show slightly higher wind speeds and temperatures compared to the ground-based instruments.

NAVGEM-HA reproduces the seasonal asymmetry of the zonal wind at mid- and high latitudes. The temperature and wind fields in NAVGEM-HA are realistic compared to ground-based sensors up to an altitude of 90 km (geometric altitude). However, our comparison also confirms that the availability of satellites observations for the data assimilation in NAVGEM-HA has an impact on the overall agreement. Further, the meteorological analysis reflects the seasonal phase behavior of the semidiurnal tide, which is constantly changing. These continuous phase changes are important and need to be considered when analyzing satellite observations or spectral analysis using long windows. NAVGEM-HA reflects the day-to-day variability of the wind and semidiurnal tide amplitude and phase behavior during SSW events. The combination of NAVGEM-HA meteorological analysis data and ground-based observations allowed us to develop new diagnostics to retrieve atmospheric information and to investigate physical processes. The cross-validation suggests that the global fields of NAVGEM-HA provide a realistic boundary to nudge other GCMs coupling the middle atmosphere to the upper atmosphere. In particular, the good agreement of the tidal phases is an essential quality benchmark for the lower forcing of the thermosphere and ionosphere through atmospheric tides. The day-to-day tidal variability (amplitude and phase) of the semidiurnal tide is associated to changes in the wind pattern in the middle atmosphere altering the vertical propagation conditions of the tide. This is in agreement with previous studies by Fuller-Rowell et al. (2010); Jin et al. (2012); Fuller-Rowell et al. (2016). Further, we did investigate a potential lunar tide amplification through the Pekeris resonance effect as proposed by Forbes and Zhang (2012); Zhang and Forbes (2014). The ASF and holographic analysis permit to determine the different phases of the SSW 2012/13 and the tidal response in much more detail with respect to the timing. The tidal enhancement after the SSW, which was in many previous studies termed to be a lunar tidal enhancement, shows essentially a period around 12.0 h (see holograms) and has the same vertical wavelength of about 50-60 km than the semidiurnal tide before the SSW. The global diagnostic also confirms a wavenumber 2 structure. Further, there are no signs of a coupling and phase relation to the lunar orbit during this time. However, during the SSW there occurs a phase shift of the semidiurnal tide towards the 12.42 h period and the dynamical and thermal structure could be suitable to shift the Pekeris resonance towards a period of 12.42 h as well, as outlined in Forbes and Zhang (2012); Zhang and Forbes (2014). The increased vertical wavelength of about 150-400 km and the time span of 3-5 days during this phase of the SSW might be the result of the resonance and may indicate the presence of a lunar tidal mode, but this needs to be confirmed by tidal modelling and is beyond the discussion herein. However, the amplitude of this tidal mode is still much smaller than a typical semidiurnal tide, but might be larger than the average lunar amplitude of about 1-4 m/s (Sandford et al., 2006). Holographic analysis provide a new method to investigate the frequency behavior using short windows in the time domain, but keeping a localized measurement of the frequency resolution. Further, we were able to provide a quantitative spectral measurement of the spectral variability for the semidiurnal tide, which pointed out that the lunar tide (M2) lies well withing the spectral line shape. This has now some implications for epoch analysis of the lunar tide from local observations. The holograms show that there are frequently shifts of the semidiurnal tide towards the M2 (12.42 h) that are disconnected to the lunar orbit, which means the lunar tide can hardly be inferred from such an analysis without additional information, for instance, the vertical wavelength of the lunar tide. In this work we have demonstrated the value of meteorological analysis data from NAVGEM-HA to investigate the day-to-day variability of tides in a global context and for local meteor radar observations. Such data sets are essential for nudging thermospheric and ionospheric models for space weather applications. Further, we emphasized that new

analysis techniques are required to infer the tidal variability or to separate lunar tides from the semidiurnal tide. Holographic reconstructions and spectral line models for atmospheric tides might be part of such a solution.

*Data availability.* The meteor radar can be obtained from Gunter Stober upon request from the Leibniz-Institute for Atmospheric Physics.
5 The NAVGEM-HA data used in this study can be obtained from (https://map.nrl.navy.mil/map/pub/nrl/navgem/iap). The lidar observations are available upon request from Kathrin Baumgarten.

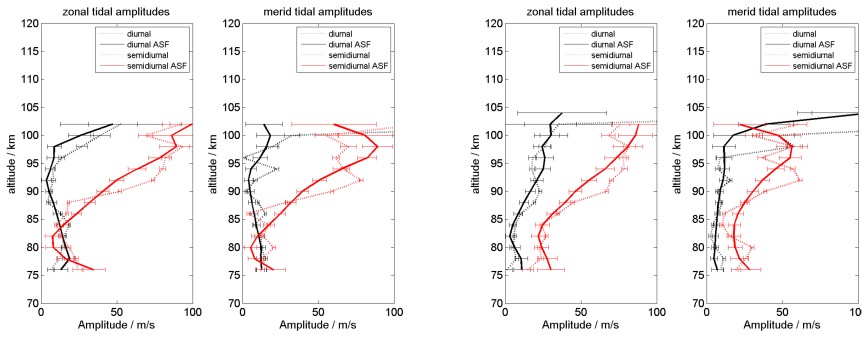

**Figure A1.** Here we show observations from 1st February 2010 and the Juliusruh meteor radar. The dashed lines indicate the tidal solution applying only temporal fitting and solid line shows the ASF solution with vertical regularization for the diurnal and semidiurnal tide.

### Appendix A:  Comparison of ASF with and without vertical regularization

Here we provide two examples comparing a tidal amplitude fit for the zonal and meridional component using the 1D ASF and the 2D ASF with vertical regularization to demonstrate how a potential contamination of gravity waves with short vertical wavelengths is reduced. The time difference between the left two panels and the right two panels is 6 hours.

### Appendix B:  Tidal components from global NAVGEM-HA analyzed winds

In addition to the semidiurnal tide locally observed from the meteor radar as well as from NAVGEM analyzed winds here we provide the results for the westward- and eastward-propagating non-migrating semidiurnal tidal components (SW1, SW3, SE1, SE2, SE3) as well as for the stationary semidiurnal tide (S0) during the winter 2009/2010 and 2012/2013 for the stations at Andenes and Juliusruh from the global fields of NAVGEM-HA.

### Appendix C:  Vertical wavelength from MR and NAVGEM-HA

Vertical wavelengths were derived from the vertical profiles of the phases of the semidiurnal tidal fit for every day. In the case of the meteor radar the fit is performed at altitudes between 74-100 km. The NAVGEM-HA data was analyzed in the altitude range from 70-90 km. However, as we estimate the vertical wavelengths from a rather thin atmospheric layer at the MLT, the uncertainty of the obtained wavelengths scales with the wavelength itself. There is a tendency that the uncertainties are larger for wavelengths beyond 250 km.

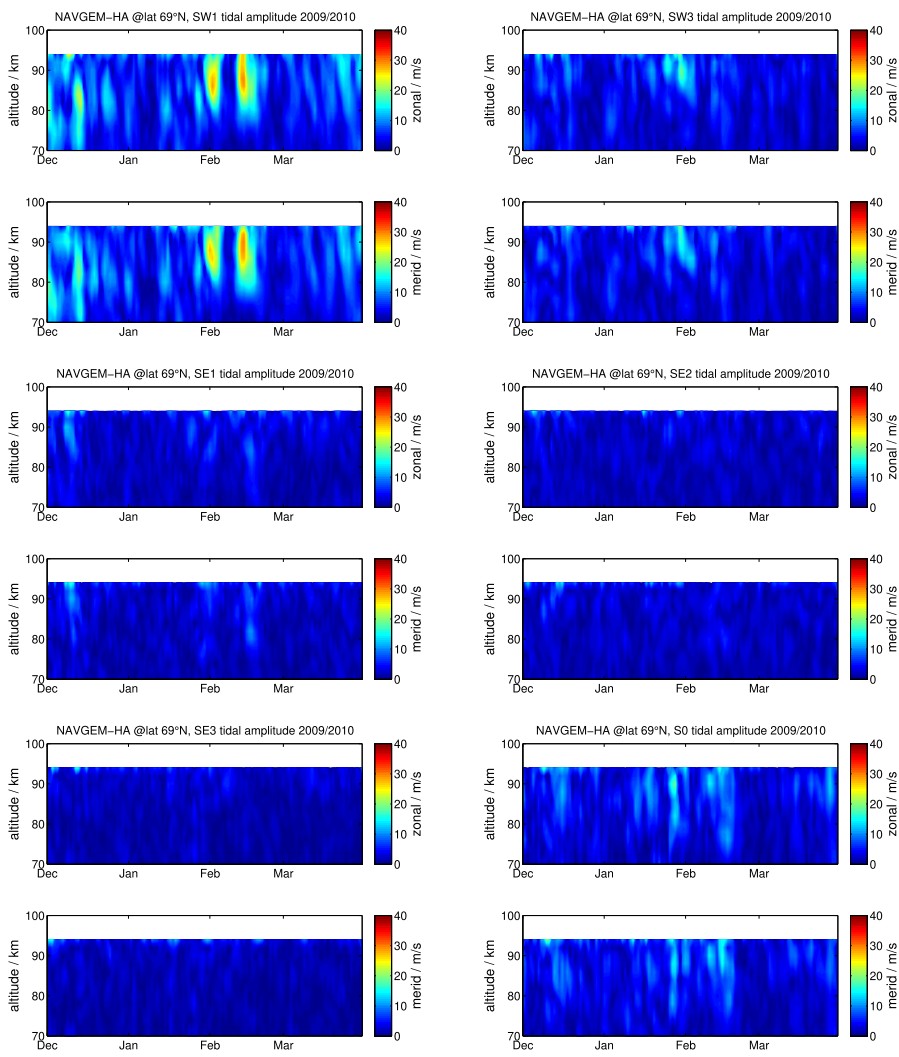

**Figure B1.** Non-migrating tides derived from global NAVGEM-HA winds above Andenes during the winter 2009/10 for SW1 and SW3 (upper two panels), SE1 and SE2 (middle panels), and SE3 and S0 (lower two panels) tidal components.

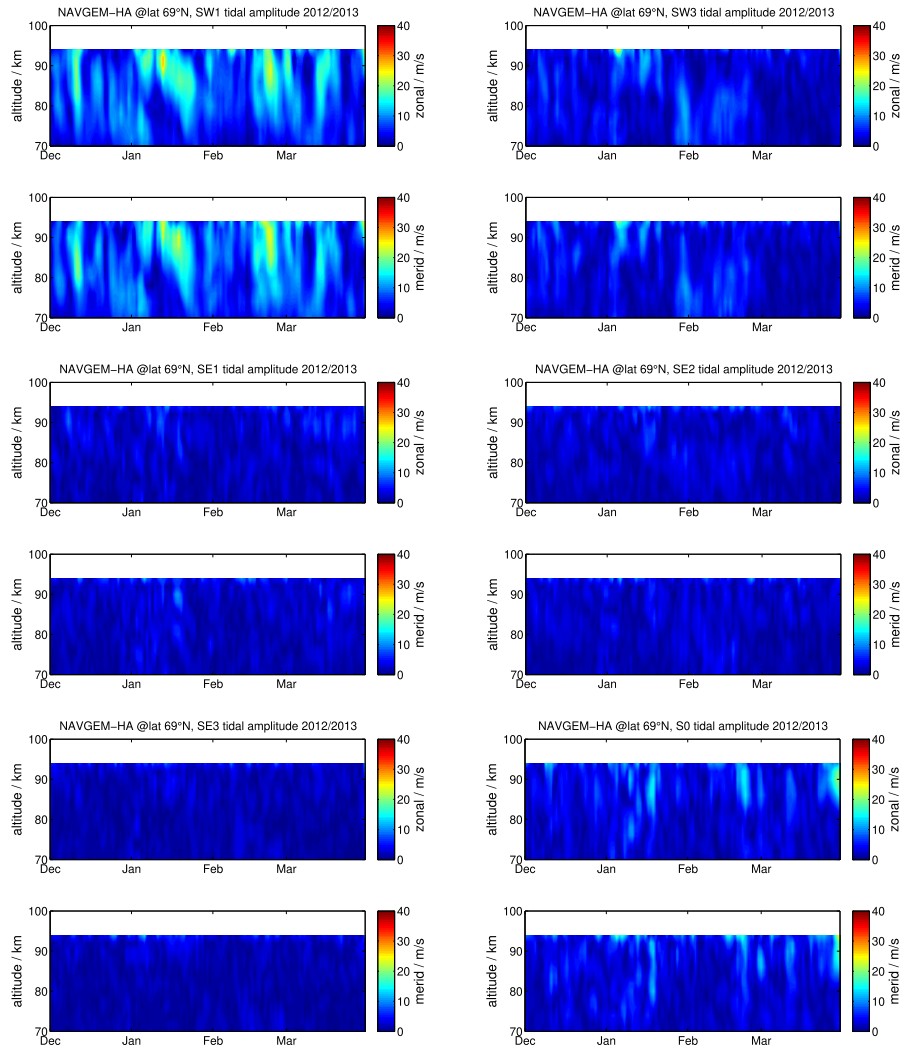

**Figure B2.** The same as Fig. 13 but for the winter 2012/2013.

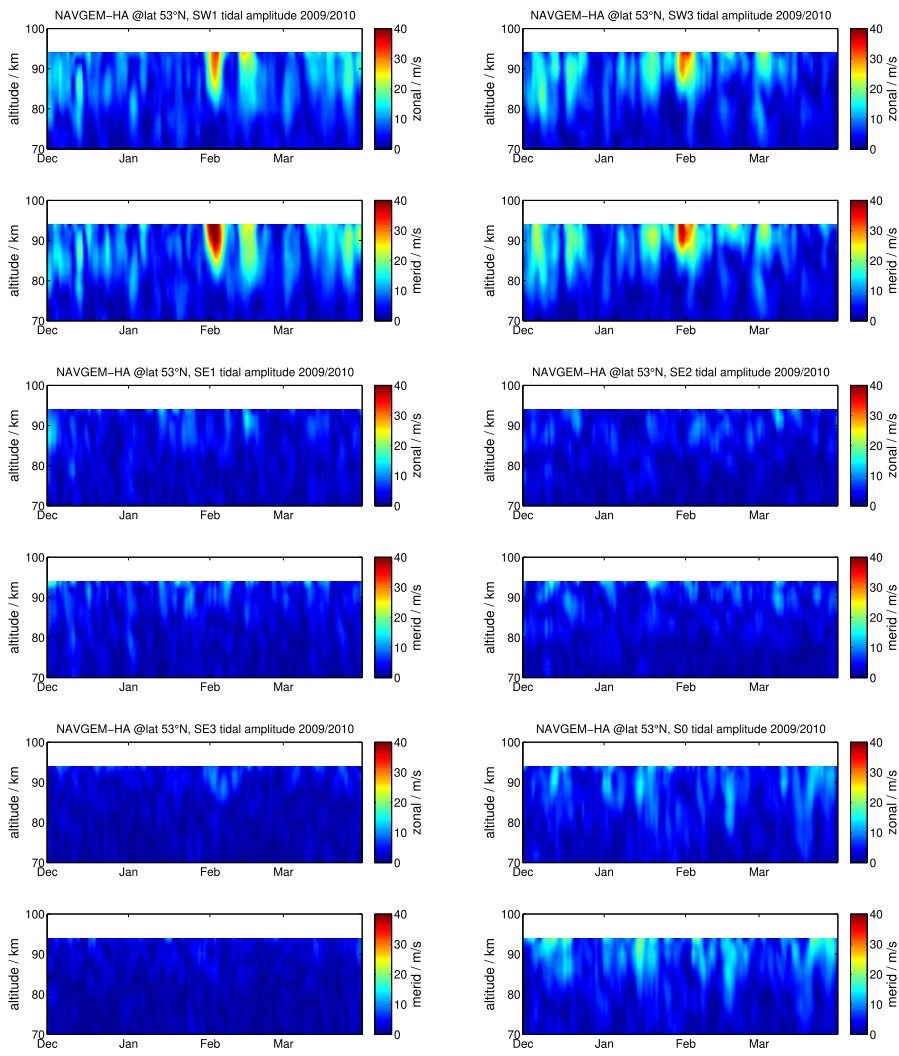

**Figure B3.** The same as Fig. 13 but above Juliusruh and for the winter 2009/2010.

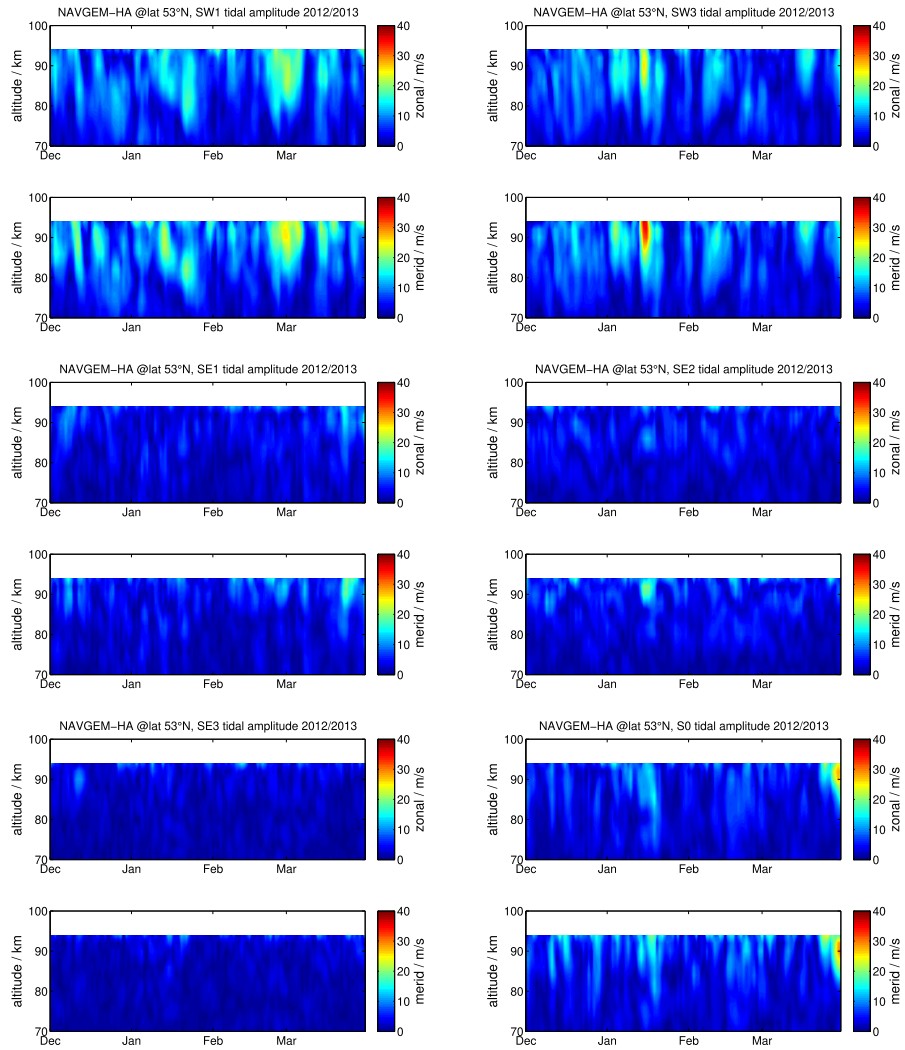

**Figure B4.** The same as Fig. 13 but above Juliusruh and for the winter 2012/2013.

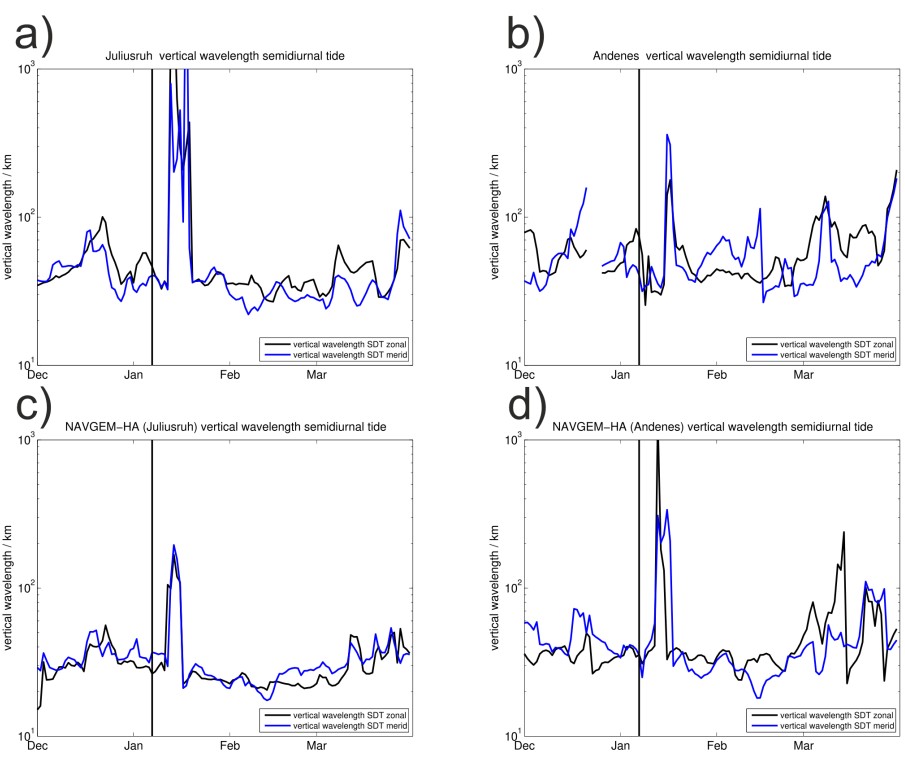

**Figure C1.** Time series of the vertical wavelength of the semidiurnal tide at Juliusruh and Andenes. The upper two panels a) and b) denote the meteor radar observations for both locations. The lower panels c) and d) are obtained from NAVGEM-HA.

*Author contributions.* The manuscript is edited and discussed with all authors. The conceptual idea of the manuscript was developed by Gunter Stober, Kathrin Baumgarten and John McCormack. The meteor radar data analysis is performed by Gunter Stober. Kathrin Baumgarten computed partly the lidar temperatures and analyzed both lidar data sets. Peter Brown contributed with the CMOR radar data, read and edited the manuscript and helped with the discussions. Jerry Czarnecki provided support in the data analysis and helped discussing the results.

*Acknowledgements.* This work is supported by the University of Bern Institute of Applied Physics and the Oeschger Center for Climate Change Research. Kathrin Baumgarten is supported by the Deutsche Forschungsgemeinschaft (DFG; German Research Foundation) under project LU1174/8-1 (PACOG) of the research unit FOR1898 within the Research Unit MS-GWaves. The work at the Naval Research Laboratory was supported by the Chief of Naval research and by a grant of computer time from the High Performance Computing Modernization Program. We gratefully acknowledge Michael Gerding, Michael Priester and Torsten Köpnick for the maintenance and operation of the lidar systems at IAP as well as all students for helping in lidar operation. We appreciate the support from Josef Höffner in computing the resonance lidar temperatures. We acknowledge the technical support of the IAP technicians in the operation of the meteor radars.

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
