# Peer review of "Comparative study between ground-based observations and NAVGEM-HA analysis data in the MLT region"

_Atmospheric Chemistry and Physics, 2019_

## Referee Comment (RC1) · Anonymous Referee #1 · 16 Dec 2019

General comments:

This study shows comparisons of MLT dynamics between the ground-based observations and the new reanalysis data which covers the mesosphere. The new analysis technique which could overcome data gap and uneven sampling in the observation is well introduced, although a setting of the vertical retrieval kernel should be carefully discussed. The authors clearly describe the good performance of NAVGEM-HA reanalysis data in terms of climatology and the short-term response to the sudden stratospheric warming. The possible mechanisms for the short-term response of the semi-diurnal tides are also well discussed in Section 5. Since this paper shows many attractive

observational/simulated results, the time-lag and/or the time scale of the short-term response of the semi-diurnal tides, in my opinion, should be a little more described in Section 4, which might be helpful for the discussion of the above mechanisms. In addition, the discussion section could be shortened by moving some sentences/paragraphs to the other sections. So, I would recommend publication of this paper only with some minor revisions described below.

Comments:

1. Page 4, line 18: It would be better to replace the sentence "The Rayleigh backscatter is ... under the assumption of hydrostatic equilibrium" by a new one; "The temperature are calculated under the assumption of hydrostatic equilibrium from the Rayleigh backscatter which is proportional to the atmospheric air density."

2. Page 4, line 22: "only down to" → "only above" ??

3. Page 4, line 29: please delete "?".

4. Page 5, line 28: What is the advantage of the ASF compared with a wavelet technique such as S transform (Stockwell et al., 1996)?

5. The benefits of the ASF and a part of the discussion for the vertical kernel described in Section 5.1 would be better to be moved in Section 2 to shorten Section 5.

6. Page 6, lines 1-12: Please insert two references about gravity waves in MLT regions: Chen et al. (2013) to (Page 6, line 9), which shows a case study of observed gravity waves with the vertical wavelength of 22~23 km. Shibuya et al. (2017) to (Page 6, line 6). which shows a case study of gravity waves with the wave periods of quasi-12 h (The climatological study of the above cases is discussed in Chen et al., 2016, JGR and Shibuya and Sato, 2019, ACP, respectively, which I think need not to be introduced here).

7. Page 7, lines 22: The altitudes of the wind reversal are quite different from the observations and the reanalysis data, which should be mention in the main text. The

altitude of the wind reversal is quite important for the breaking condition of the upward propagating gravity waves.

8. Page 8, lines 9: Why is the amplitude of the semi-diurnal tides in reanalysis data overestimated above the altitude of 90 km? I'm afraid that this point is not discussed in Section 5.

9. Page 10, in Figure 3: Please add the explanation to the representation of a tidal phase (p12?).

10. Page 10, line 9: Please mark the central date of the sudden stratospheric warming in the figures after Fig. 6.

11. Page 10, line 12: Why does the data gap in the observation at Andenes exist near the central date of SSW? Is this related to the SSW?

12. Page 11, line 6 (CRITICAL): Please mention the time-lag between the central date of the SSW and the amplification of the semidiurnal tide both in the observation and the reanalysis data in Figs 6, 7, 9 and 10, respectively.

13. Page 12, line 4: In Figure 8, the SW2 tidal amplitude seems to decrease after the central date of SSW below the altitude of 85 km? Such a decrease is not dominant in each localized point in NAVGEM-HA in Figs. 6 and 7. Why is this found only in the zonal mean?

14. Page 12, line 21-24: Please move the sentence "Atmospheric ..." to Introduction.

15. Page 22, line 4-9: For the discussion of the amplification of the tides after the SSW, the time-lag of the amplification should be one of the key components. For example, the time-lag might be related to the vertical group velocity of the tides which propagate from the source region. Did the previous study discuss such a time-lag in their proposed mechanism?

16. Page 22, line 24: Moreoverr→Moverover.

References: Chen, C., Chu, X., McDonald, A. J., Vadas, S. L., Yu, Z., Fong, W., and Lu, X.: Inertia‐gravity waves in Antarctica: A case study using simultaneous lidar and radar measurements at McMurdo/Scott Base (77.8° S, 166.7° E). Journal of Geophysical Research: Atmospheres, 118(7), 2794-2808, 2013. Stockwell, RG; Mansinha, L; Lowe, RP (1996). "Localization of the complex spectrum: the S transform". IEEE Transactions on Signal Processing. 44 (4): 998–1001. CiteSeerX 10.1.1.462.1500. doi:10.1109/78.492555. Shibuya, R., Sato, K., Tsutsumi, M., Sato, T., Tomikawa, Y.,Nishimura, K., and Kohma, M.: Quasi-12 h inertia–gravity waves in the lower mesosphere observed by the PANSY radar at Syowa Station (39.6_ E, 69.0_ S), Atmos. Chem. Phys., 17, 6455–6476, 2017

---

## Referee Comment (RC2) · Anonymous Referee #2 · 23 Dec 2019

Comparative study between ground-based observations and NAVGEM-HA reanalysis data in the MLT region G. Stober et al.

The authors present a study of tidal variability at altitudes of 75 to 110 km in the northern mid-to-high latitudes, emphasizing periods around stratospheric sudden warmings. They compare observations from a number of sources, most notably meteor radars, to NAVGEM-HA analyses. They also present a diagnostic tool called an adaptive spectral filter. It is not clear to me what the central purpose of the study is: is this a validation of the NAVGEM-HA reanalysis? is this a methodology paper introducing the adaptive spectral filter? Is this a science paper focusing on the variability of the tides around

sudden warmings? I am not sure what the reader is supposed to take away from this paper

One symptom of this is that the bullet-point list of conclusions in the final section is vague. Several bullet points claim that the reanalyses are 'realistic' and suitable for use as lower boundary conditions, but criteria for this claim are never discussed, and other validaion papers cited seem to have drawn these conclusions already. Variations in the tide are attributed to variations in the 'wind patterns' of the middle atmosphere but no evidence is provided to support this claim. The merits of the ASF methodology (e.g. error estimates) are touted but never used. And another 'holographic reconstruction' methodology is used in the discussion section without ever being introduced.

I find the figures difficult to read, numerous, and not clearly organized with respect to the discussion, again my sense is that this is a symptom of the paper not having a clear purpose. Finally, the text of the manuscript is still rough around the edges, with incomplete senteces and missing references.

I've included a list of specific comments below. On the basis of the above comments it is my opinion that this manuscript should be substantially revised before it can be considered suitable for publication.

Specific Comments:

Section 2: Data and Model output:

The periods for which data are available for each data source are not given. Neither are the 'analyzed periods' specified.

To what end have the temperature observations been included?

Are the NAVGEM-HA outputs analyses or reanalyses? To what extent should the reader expect the tidal structures analyzed in this paper to be directly constrained by assimilated observations?

What kind of sponge layer does the forecast model employ and over what levels does it act?

There are missing citations in the first and third paragraph of section 2.3.

Section 3: Diagnostics

The details of the ASF are vague to the point that it is difficult to assess the validity of any of the results. For instance: how is the sliding window determined? What windows are in fact used? What is the purpose of the scaling factor? How is the vertical 'regularization' carried out? If these details are given in previous studies this should be clearly stated, if they are novel they should be justified. No specifics are given about how planetary-scale waves are accounted.

Details of the 'holographic reconstruction' methodology discussed in Fig. 12 should be given in this section.

Section 4: Results

Figure 1: What time periods have been used to create these figures?

My reading of the figures is that the summertime reversal of the zonal winds from easterly to westerly occurs at higher altitudes in NAVGEM analyses than in the radar data, and that the southward meridional winds are not as strong. Is this the bias that is reiterated in the conclusions? Has this bias been noted in previous work?

Figure 2: The warm anomalies in NAVGEM-HA near 95 km are plausibly a sponge layer effect - one would need to know details of the sponge layer to assess this claim.

Figs. 3 to 5: The structure of the discussion (which discusses first observations then NAVGEM-HA) does not match the structure of the figures. More importantly the tidal amplitudes and phases in NAVGEM-HA do not look like close matches to observations to my mind. This would be a useful place to make use of the error propgation capabilities of the ASF methodology that are claimed as a benefit in later discussion.

Figs 4 to 6, 7 to 9: Again the structure of the figures and the discussion don't match up.

Section 5:

The merits of error propagation through the ASF methodology has not been demonstrated, nor has the benefits of the vertical resolution. I can see that these are both desirable features but no demonstration has been made of their value or correctness.

p22 lines 33-35: SSWs can perturb the middle atmosphere for months, as was the case in both the 2008-9 and 2012-3 events considered here.

Figure 12: What is the difference between the upper and lower panels? Also, the units for the period are wrong.

---

## Author Comment (AC1) · 27 Jan 2020

Reply to Reviewer:

General comments: This study shows comparisons of MLT dynamics between the ground-based observations and the new reanalysis data which covers the mesosphere. The new analysis technique which could overcome the data gap and uneven sampling in the observation is well introduced, although a setting of the vertical retrieval kernel should be carefully discussed. The authors clearly describe the good performance of NAVGEM-HA reanalysis data in terms of climatology and the short-term response to the sudden stratospheric warming. The possible mechanisms for the short-term

response of the semi-diurnal tides are also well discussed in Section 5. Since this paper shows many attractive observational/simulated results, the time-lag and/or the time scale of the short-term response of the semi-diurnal tides, in my opinion, should be a little more described in Section 4, which might be helpful for the discussion of the above mechanisms. In addition, the discussion section could be shortened by moving some sentences/paragraphs to the other sections. So, I would recommend publication of this paper only with some minor revisions described below.

General reply: We thank the reviewer for his constructive and helpful comments about the submitted manuscript. We revised the manuscript according to his suggestions. However, as the second reviewer recommended a more extensive revision, the changes to the manuscript are in some paragraphs substantial. A point by point reply to each raised comment is given below.

Comment: 1. Page 4, line 18: It would be better to replace the sentence "The Rayleigh backscatter is...under the assumption of hydrostatic equilibrium" by a new one; "The temperature is calculated under the assumption of hydrostatic equilibrium from the Rayleigh backscatter which is proportional to the atmospheric air density."

Reply: (Page: 5 line: 3) We changed the sentence as suggested.

Comment: 2. Page 4, line 22: "only down to"→"only above" ??

Reply: (Page: 5 line: 6) We followed the suggestion.

Comment: 3. Page 4, line 29: please delete "?".

Reply: (Page: 5 line: 14) There was a reference missing due to a mistake in the latex file. We cited Kuhl et la., 2013.

Comment: 4. Page 5, line 28: What is the advantage of the ASF compared with a wavelet technique such as S transform (Stockwell et al., 1996)?

Reply: (Page: 7 line: 29-) We added a short paragraph discussing the pro and

cons of both techniques. The ASF technique aims, similar to the S-transform \citep{Stockwell:1996}, to infer spectral information of intermittent signals. However, the S-transform is based on wavelet techniques and, thus, takes all the pros and cons of these methods. The main benefits of the ASF are given in the error, the possibility to use unevenly sampled time series with data gaps and most importantly to apply individual constraints (in this study vertical wavelengths) to each fitted frequency component. Both methods should yield similar results for model data sets that obey the requirements mentioned above.

Comment: 5. The benefits of the ASF and a part of the discussion for the vertical kernel described in Section 5.1 would be better to be moved in Section 2 to shorten Section 5.

Reply: (Page: 7 line: 21-28) We moved parts of section 5.1 to the ASF section and linked the new paragraph to this discussion.

Comment: 6. Page 6, lines 1-12: Please insert two references about gravity waves in MLT regions: Chen et al. (2013) to (Page 6, line 9), which shows a case study of observed gravity waves with the vertical wavelength of 22âĹij23 km. Shibuya et al. (2017) to (Page 6,line 6). which shows a case study of gravity waves with the wave periods of quasi-12h (The climatological study of the above cases is discussed in Chen et al., 2016, JGR and Shibuya and Sato, 2019, ACP, respectively, which I think need not to be introduced here).

Reply: (Page: 7 line: 13-16) We added a short discussion of the first two publications into the paragraph. Both publications are interesting and highly relevant for this study. As the rather long vertical and horizontal wavelength, which are reported in both papers at such high latitudes brings new issues to the debate on how to separate a tide from a gravity wave at the polar regions. Further, considering She et al., 2016, who outlined that tidal waves satisfy the polarization relation for gravity waves.

Comment: 7. Page 7, lines 22: The altitudes of the wind reversal are quite different

from the observations and the reanalysis data, which should be mention in the main text. The altitude of the wind reversal is quite important for the breaking condition of the upward propagating gravity waves.

Reply: (Page: 7 line: 13-16) We added some sentences explicitly pointing at these differences at their relevance for gravity wave propagation and breaking. On the other side, we have to mention that NAVGEM-HA winds and temperature fields are in much better agreement with the observation than many other GCM's perform at these altitudes. Even gravity wave resolving models seem to have difficulties to reproduce the observations in such details. Maybe such comparisons should become a benchmark to validate and cross-compare in climatological sense free-running GCM's, reanalysis data sets and meteorological analysis (e.g. NAVGEM-HA).

Comment: 8. Page 8, lines 9: Why is the amplitude of the semi-diurnal tides in reanalysis data overestimated above the altitude of 90 km? I'm afraid that this point is not discussed in Section 5.

Reply: (Page: 5 line: 18-28) We added a paragraph outlining the issue with the altitudes above 90 km. After removing the sponge layer from NAVGEM-HA and converting the geopotential altitudes to geodetic altitudes using WGS84 the uppermost trustworthy altitude is 92 km during winter and 90 km during summer. In fact, tidal amplitudes should not be interpreted beyond these altitudes. Our regridding up to 94 km led to an extrapolation of the tidal amplitudes, which was further enhanced due to the vertical regularization of the ASF. This is now clearly stated in the manuscript.

Comment: 9. Page 10, in Figure 3: Please add the explanation to the representation of a tidal phase (p12?).

Reply: (Figures 3,4,5,6,7 and 9,10) We added an explanation of the labels to the figure caption.

Comment: 10. Page 10, line 9: Please mark the central date of the sudden strato-

spheric warming in the figures after Fig. 6.

Reply: (Figures 3,4,5,6,7,8 and 9,10,11,12,13) We indicated the onset of the SSW for each figure using the definition from McCormack et al., 2017. The onset is given by a black vertical line.

Comment: 11. Page 10, line 12: Why does the data gap in the observation at Andenes exist near the central date of SSW? Is this related to the SSW?

Reply: The Andenes MR radar had a technical problem and was off for some days. This happens frequently. Mostly due to the icing of the antennas or strong winds, which significantly degrades the VSWR and triggers a shut-down of the transmitter. Whether this was related to the SSW is beyond our knowledge.

Comment: 12. Page 11, line 6 (CRITICAL): Please mention the time-lag between the central date of the SSW and the amplification of the semidiurnal tide both in the observation and the reanalysis data in Figs 6, 7, 9 and 10, respectively.

Reply: We agree to the reviewer that the time-lag between the SSW and the onset of the enhancement of the semidiurnal tide is important. We are already preparing another study with more events to systematically look at this pattern and time scales. However, as the vertical propagation of the semidiurnal tide is mostly affected by the local(regional) air packages in the column around our measurement locations, the classical definition of a central day of an SSW seems to be not appropriate to measure the time-lag (e.g. the zonal wind reversal at 60°N). For different latitudes, the zonal wind and the zonal wind reversal depend on the polar vortex position and its evolution, thus, we have to find another definition to measure the time lag. If we define the max of the zonal wind reversal at a given latitude at 70 km altitude as central day, the time-lag is about 1-3 days. It takes 1 day for the onset of the tide amplification and 2-3 days to reach the maximum amplitude. If we use the standard definition of the central day at 10 hPa and at 60°N, the time lag is between 3-6 days at Andenes and 2-3 days at Juliusruh. Thus, the discussion of time delays and a potentially new definition of the central

day at local coordinates would require a more detailed study, which is in preparation.

Comment: 13. Page 12, line 4: In Figure 8, the SW2 tidal amplitude seems to decrease after the central date of SSW below the altitude of 85 km? Such a decrease is not dominant in each localized point in NAVGEM-HA in Figs. 6 and 7. Why is this found only in the zonal mean?

Reply: We were not aware of this feature so far. A detailed analysis is beyond the scope of the paper. However, there are two aspects that are relevant and need to be further disentangled. As shown in the appendix, there occur short and sudden enhancement of non-migrating tides before and after the SSW event (SW1, SW3), which might just be an artifact due to aliasing or a real excitation of both non-migrating tides modes. The local diagnostic only reveals a superposition of the migrating and non-migrating modes and, thus, depending on the phase behavior and the longitude of the observations, they might pick up only the positive interference of all tidal modes. The second aspect is the planetary waves and how the SSW affects the polar vortex. In the case of the SSW 2010, the polar vortex was clearly displaced to the European sector (see publications of Kodera et al., 2016), which is the sector of our observations. As a result, the local diagnostic can look rather different with respect to the zonal mean. The planetary wave also has an effect on the amplitude of the tides can grow with altitude. Depending on the PW structure of wave numbers 1,2 and 3 the vertical propagation of tides is affected.

Comment: 14. Page 21, line 21-24: Please move the sentence "Atmospheric..." to Introduction.

Reply: (Page:3 line:17-29) We moved the sentence to the introduction.

Comment: 15. Page 22, line 4-9: For the discussion of the amplification of the tides after the SSW, the time-lag of the amplification should be one of the key components. For example, the time-lag might be related to the vertical group velocity of the tides which propagate from the source region. Did the previous study discuss such a timelag in their proposed mechanism?

Reply: We thank the reviewer for making this comment. The vertical propagation of the tides is indeed a key element and, thus, the time lag between the SSW and the enhancement at least not in the context of the lunar tide. Only Forbes and Zhang (2012) mentioned the time delay between the central day and the semidiurnal tide amplification, which they then attributed to a lunar tide. However, given the dramatic change in the vertical wavelength of the semidiurnal tide from 50-60 km before the SSW to 200-300 km during a back after 3-5 days to 50/60 km indicates already that the vertical group speed is essential. We currently working on a more detailed analysis using a more extended dataset of NAVGEM-HA and observations. We now put more emphasis on this aspect throughout the manuscript. However, a more detailed study is in preparation for more data and events.

Comment: 16. Page 22, line 24: Moreoverr→Moverover.

Reply: (Page: 25 line:22) Done.

Comment: References: Chen, C., Chu, X., McDonald, A. J., Vadas, S. L., Yu, Z., Fong, W., and Lu, X.: Inertia gravity waves in Antarctica: A case study using simultaneous lidar and radar measurements at McMurdo/Scott Base (77.8âŮęS, 166.7âŮęE). Journal of Geo-physical Research: Atmospheres, 118(7), 2794-2808, 2013. Shibuya, R., Sato, K., Tsutsumi, M., Sato, T., Tomikawa,Y.,Nishimura, K., and Kohma, M.: Quasi-12 h inertia–gravity waves in the lower meso-sphere observed by the PANSY radar at Syowa Station (39.6_ E, 69.0_ S), Atmos. Chem. Phys., 17, 6455–6476, 2017

Reply: We thank the reviewer for providing these additional references and included them at the suggested paragraph in the manuscript.

———————————————————

---

## Author Comment (AC2) · 27 Jan 2020

Comparative study between ground-based observations and NAVGEM-HA reanalysis data in the MLT region G. Stober et al.The authors present a study of tidal variability at altitudes of 75 to 110 km in the north-ern mid-to-high latitudes, emphasizing periods around stratospheric sudden warmings. They compare observations from a number of sources, most notably meteor radars, to NAVGEM-HA analyses. They also present a diagnostic tool called an adaptive spectral filter. It is not clear to me what the central purpose of the study is: is this a validation of the NAVGEM-HA reanalysis? is this a methodology paper introducing the adaptive spectral filter? Is this a science paper focusing on the variability of the tides around sudden warmings? I am not sure what the reader is supposed to take away from this paper

One symptom of this is that the bullet-point list of conclusions in the final section is vague. Several bullet points claim that the reanalyses are 'realistic' and suitable for use as lower boundary conditions, but criteria for this claim are never discussed, and other validation papers cited seem to have drawn these conclusions already. Variations in the tide are attributed to variations in the 'wind patterns' of the middle atmosphere but no evidence is provided to support this claim. The merits of the ASF methodology (e.g. error estimates) are touted but never used. And another 'holographic reconstruction' methodology is used in the discussion section without ever being introduced.

I find the figures difficult to read, numerous, and not clearly organized with respect to the discussion, again my sense is that this is a symptom of the paper not having a clear purpose. Finally, the text of the manuscript is still rough around the edges, with incomplete sentences and missing references.

I've included a list of specific comments below. On the basis of the above comments it is my opinion that this manuscript should be substantially revised before it can be considered suitable for publication.

**General Reply:**

We thank the reviewer for his constructive comments to the submitted paper. We have revised the manuscript according to his suggestions and included new and the missing citations, added paragraphs providing some of the suggested information and restructured parts of the manuscript to provide a more consistent narrative.

NAVGEM-HA has not yet been validated in the climatological sense using independent ground base sensors. It is not worth to investigate the short time variability, if the seasonal climatology is not well-reproduced. The short-term variability and cross-validation of NAVGEM-HA fields with respect to specific waves is a new way to benchmark meteorological analysis systems, but is also

tied to the methodology to extract the information, which is in the case the ASF technique. Although, we don't intent to focus on the method itself, it is necessary to provide essential information on the technique.

Finally, we present a detailed discussing of the tidal variability related to a highly relevant coupling process at the middle atmosphere (SSW) and its relation to alter in this case the semidiurnal tide. We discuss potential affects in the context of lunar tides, which provides an excellent example and demonstrate the potential of combining various local and global data sets to analysis effects on time scales that are hardly accessible with other methods at MLT altitudes.

In so far, we want to keep the general content of the paper, but did, as suggested, revise the structure and moved some paragraphs to get a better structure. We hope that the revised version satisfies the reviewers suggestions and comments.

Detailed answers are provided below for each comment.

Later there will be a tracked changes file uploaded. The red color labels removals, the blue color insertions.

Specific Comments:

Section 2:

Data and Model output:

**Comment:**

The periods for which data are available for each data source are not given. Neither are the 'analyzed periods' specified.

**Reply:** (Page: 3 line:3-4)

This information is provided in the section about mean winds.

**Comment:**

To what end have the temperature observations been included?

**Reply:**

There are only a few ground based temperature climatologies available. So far NAVGEM-HA was not yet compared to independent temperature observations. Satellite temperature measurements from MLS and SABER don't provide an independent data set due to the assimilation.

**Comment:**

Are the NAVGEM-HA outputs analyses or reanalyses? To what extent should the reader expect the tidal structures analyzed in this paper to be directly constrained by assimilated observations?

**Reply:**

NAVGEM-HA is a meteorological analysis, we removed through the manuscript the term reanalysis.

**Comment:**

What kind of sponge layer does the forecast model employ and over what levels does it act?

**Reply**: (Page: 5 line: 18-28)

We provide this information in section about NAVGEM-HA.

**Comment:**

There are missing citations in the first and third paragraph of section 2.3.

**Reply:**

The missing reference is now added.

Section 3: Diagnostics

**Comment:**

The details of the ASF are vague to the point that it is difficult to assess the validity of any of the results. For instance: how is the sliding window determined? What windows are in fact used? What is the purpose of the scaling factor? How is the vertical 'regularization' carried out? If these details are given in previous studies this should be clearly stated, if they are novel they should be justified. No specifics are given about how planetary-scale waves are accounted.

**Reply:** (Page: 31 appendix A1)

We added the reference of previous developments of the ASF technique and how it was validated. Here we just mention the most important information of what was used in the ASF analysis. Further, we added a sentence outlined in more detail how the scaling factor is used to determine the window length.

We also add some figures in the appendix outlining how the technique works and about the error statistics. However, it will be critical for the readability of the

submitted paper, if much more details about the ASF implementation are added to the paper.

The implementation of the ASF is in Fortran 77 and Matlab based on modified numerical recipes algorithms. Basically we generate for all times and altitudes Jacobian matrices, which can be written as one large block diagonal matrix (in development) or we keep each Jacobian (this is the current version) and create a cubic tensor (for each tidal frequency a separate layer due to the different window lengths). Then we solve each Jacobian and store the solutions into a vector. First for the diurnal tide. These solutions are used as regularization for the next layer with the Jacobian of the semidiurnal tide and so forth. Finally, we select all altitudes falling into the vertical averaging kernel and perform a weighted linear fit to all coefficients.

Below are two examples from the same day in 01$^{st}$ February 2010 observed at Juliusruh to visualize how the ASF reduce a potential contamination due to gravity waves with short vertical wavelengths. Filtering just in time domain would move energy from such gravity waves to the tidal energy budget. The plots also indicate that at the upper and lower edges the errors get very large, which is expected as we have no longer enough measurements to perform a statistical reliable regularization. The plots contain just the profiles obtained at a specific time at the day without temporal averaging in the case of the ASF. Therefore, the errorbars are scaled by 1/sqrt(n) to make them comparable to the temporal averaged values plotted as dashed line.

[Figure]

[Figure]

**Comment:**

Details of the 'holographic reconstruction' methodology discussed in Fig. 12 should begiven in this section.

**Reply:** (Page: 25 line:2-16)

Holographic techniques are standard physical methods to derive radar parameters. However, we agree to the reviewer that it might be better to include a short paragraph describing how the hologram is obtained and used here in.

Section 4: Results

**Comment:**

Figure 1: What time periods have been used to create these figures? My reading of the figures is that the summertime reversal of the zonal winds from easterly to westerly occurs at higher altitudes in NAVGEM analyses than in the radar data, and that the southward meridional winds are not as strong. Is this the bias that is reiterated in the conclusions? Has this bias been noted in previous work?

**Reply:** (Page: 3 line:3-4)

We expanded the description of NAVGEM-HA, as the reviewer had already suggested this in the data and model section. This comparison is one of the first ones using summer-time NAVGEM-HA data for a comparison with independent ground based observations. The systematic differences in magnitude were not yet reported in a similar way in previous studies, although they were present there as well, but less obvious.

**Comment:**

Figure 2: The warm anomalies in NAVGEM-HA near 95 km are plausibly a sponge layer effect - one would need to know details of the sponge layer to assess this claim.

**Reply:** (Page: 5-6, lines 27-6(next page))

This point is also related to the model description section. We clearly remark to not use the summer-time data above 90 km due to sponge layer and extrapolation effects. The uppermost recommended usable pressure level in NAVGEM-HA after removing the sponge layer corresponds to a geometric altitude of 89/90 km at high and middle latitudes for the summer months. For the winter months the geometric altitude is between 91/92 km. Just focusing on the altitudes below 90 km, the agreement is much more reasonable compared to many other GCM's.

**Comment:**

Figs. 3 to 5: The structure of the discussion (which discusses first observations then NAVGEM-HA) does not match the structure of the figures. More importantly the tidal amplitudes and phases in NAVGEM-HA do not look like close matches to observations to my mind. This would be a useful place to make use of the error propagation capabilities of the ASF methodology that are claimed as a benefit in later discussion.

**Reply:** (Page:5 line:18-28)

We moved some paragraphs sections to the model and data description sections to get better structure of the manuscript. Due to the required large size of the figures there is a clear mismatch between figure positions and text, however, this is very difficult to be fixed in the draft stage.

The mismatch of the tidal phases and amplitudes in NAVGEM-HA and the meteor radar at altitudes above 90 km is attributed to sponge layer and extrapolation effects during the gridding to geometric altitudes.

**Comment:**

Figs 4 to 6, 7 to 9: Again the structure of the figures and the discussion don't match up.

**Reply:**

Due to the restructuring of some paragraphs this should have been improved.

Section 5:

**Comment:**

The merits of error propagation through the ASF methodology has not been demonstrated, nor has the benefits of the vertical resolution. I can see that these are both desirable features but no demonstration has been made of their value or correctness.

**Reply:**

The ASF methodology was already used in several publications before (Stober et al., 2017 (temporal ASF only), Stober et al., 2018a (gravity wave analysis using a MST radar), Stober et al., 2018b (retrieval of horizontally resolved meteor radar winds), Wilhelm et al. 2019 (mean tidal and wind climatologies as well as long term change analysis including significances based on full error propagation) and Baumgarten et al., 2019 (introduction and validation of 2D ASF using MERRA and lidar temperatures) as well as Pokhotelov et al. 2019 (cross comparison of GCM tides and meteor radar tidal climatologies). The benefit of wind retrieval errors and advanced statistical analysis including ASF filtering for spars data was demonstrated in Gudadze et al., 2019). So far substantial methodological problems were not raised by the other reviews and did not occur comparing the analysis to other climatologies.

The benefit of the error propagation is difficult to demonstrate. We propagate the error from the statistical uncertainties derived from the radar doppler velocity, which is based on the raw voltage statistics at the antennas, until the finally obtained wind or tidal component. The results presented herein are based making use of all these developments and we are sure there are differences, if we would redo all the analysis without such a weighting by the statistical uncertainties and the involved non-linear error models. If the results could be obtained without all the involved mathematics – there would be no benefit.

Please have a look on the following sequence of pictures. The left panel shows always the original parameter as zonal and meridional component, the right panels show the corresponding measurement uncertainties (here denoted error) in m/s.

Hourly winds computed using the algorithm presented in Stober et al., 2018.

[Figure]

Daily mean winds after decomposing the time series with the ASF.

[Figure]

Diurnal tidal component

[Figure]

**Semidiurnal component**

[Figure]

**Comparison of hourly winds and reconstructed time series from mean winds and tides**

[Figure]

The reconstructed time series captures remarkably good the intermittent behavior of the observed tides and of the background mean winds.

**Comment:**

p22 lines 33-35: SSWs can perturb the middle atmosphere for months, as was the case in both the 2008-9 and 2012-3 events considered here.

**Reply:** (Page: 26 line:10-11)

The reviewer is correct. We rephrase this sentence to avoid confusion about the seasonal impact of SSW (Baldwin and Dunkerton, 2003 and many other publications). We now state that the wind reversal and cooling at the MLT last only for a few days during a SSW.

**Comment:**

Figure 12: What is the difference between the upper and lower panels? Also, the units for the period are wrong.

**Reply:**

We correct the unit and uploaded a new Figure including a vertical line indicating the onset of the SSW using the criteria presented in McCormack et al., 2017.

---

## Author Response (AR2)

Reply to reviewer #1:

We thank the reviewer for his positive evaluation of the submitted manuscript.

Reply to reviewer #3:

General reply:

We thank the reviewer for reading the manuscript and his/her constructive comments on the content and structure of the manuscript. The other reviewer rated the manuscript as publish as is. This spread in the assessment seems to underline that the manuscript investigates an interesting and controversial topic and derives substantial conclusions.

The authors have the impression that one of the major concerns raised by the reviewer about the novelty of the submitted manuscript are related to a misunderstanding of the work published previously in McCormack et al., 2017, who presented a first analysis of the mean winds focusing only on **boreal winter** dynamics around the SSW 2009/2010 and 2012/13. In this manuscript we present the first cross-comparison of mean winds with NAVGEM-HA and ground based meteor radars as well as a lidar using a full season, which presents a novelty in this respect. Further, the applied wave decomposition with a recently developed technique that is termed ASF throughout the manuscript, impacts the seasonal analysis with respect to each wave component compared to spectral analysis such like FFT or wavelet analysis, which typically use much longer windows to filter for atmospheric waves.

A major aim of this manuscript is also the focus on the phase variability of tides on seasonal and interday time scales and its relation to atmospheric dynamics. We also introduce a holographic analysis to account for the tidal phase variations, which was not yet included in the S-transform analysis shown in McCormack et al., 2017. Although some of the data was already published in McCormack et al., 2017, we present more details and new aspects of the NAVGEM-HA data set.

As a benchmark of the ASF and holographic analysis, together with the global fields from NAVGEM-HA we investigate in detail the postulated lunar tide enhancement due to the Pekeris resonance effect. Our analysis presents entirely new aspects on such data sets can be analyzed, which is a key aspect to study such intermittent and transient effects as the Pekeris resonance.

**General comments:**
The present manuscript compared the MLT winds obtained from meteor radar measurements at three mid-to-high latitude stations and those from NAVGEM-HA for the daily-mean, semidiurnal tides and their day-to-day variability during the SSW. It is shown that the NAVGEM-HA reproduces the observed winds reasonably well.

A major concern is that the novelty is not clear. Especially, the same type of comparison has been already made extensively by McCormack et al. (2017) including semidiurnal tides and their variability

during SSW (the same cases) with the same datasets. Another major concern is that the manuscript is not well organized and sounds quite redundant for me. I sometimes get lost about what the authors are trying to suggest.

A small novelty of this study may be applying APF technique. And if so, I might recommend that the manuscript just focus on a subject of the day-to-day variability during SSW, although I do not find the relevant discussion (Sec. 5) written straightforward. For the discussion of Pekeris resonance, the authors resort on the results of Forbes and Zhang (2012); but it should be noted that the resonance period depends on the circulation pattern for each SSW and I would suspect that FZ2012 results cannot be applied so simply.

For the above reasons, I am afraid I feel this manuscript needs substantial revisions before the publication may be considered. Please see below for specific comments.

Major comments:

**Comment:**
1. I could not find what is a new finding for this study. McCormack et al. (2017) already made a comparison for the MLT winds between meteor radar measurements and NAVGEM-HA. They discussed the seasonality and day-to-day variability (during SSW) of semidiurnal tides. The present manuscript seems to emphasize a priority of ASF technique, but I do NOT think this technique is necessary at least for the analysis of seasonality (the variation with a relatively long time scale). In this context, I would barely guess that the discussion in Sec. 5.4 seems scientifically interesting; if so I might recommend the manuscript just concentrate on this subject, substantially reducing the rest part.

**Reply:**

McCormack et al., 2017 showed initial results of two **boreal winter seasons** of NAVGEM-HA and compare these three months with worldwide distributed meteor radars. The seasonality **was not included or discussed** given that there was no NAVGEM-HA data for a full year available at the time. Here we present the first time a seasonal cross-comparison of NAVGEM-HA and ground-based observations. However, for the sake of completeness the seasonal comparison also includes NAVGEM-HA outputs that was presented in McCormack et al., 2017. Further, the submitted manuscript emphasis the phase variability of tides and different tidal modes, which was not covered previously.

**Comment:**
2. Partly because of the issue #1, the manuscript is not well organized. Particularly I think Section 5 should only focus on scientific discussions about the results shown in Section 4. Sec. 5.1 should be briefly merged into Sec. 3 and Section 5.2 be into Sec. 2.3, because they are all related to the data quality and analysis methods. Section 5.3 can be entirely removed, because the description here is like an Introduction of tides and a brief summary of results, not providing any detailed scientific discussions.

**Reply:**

As suggested by the reviewer we restructured parts of the discussions as suggested. Thus, we removed some redundant paragraphs from the discussion or merged them with section 2 and 3. All paragraphs discussing the data quality and analysis are moved to the suggested sections. Repetitions were removed. The discussion of the mean winds and tide seasonal climatology is now more focused. We have to mention that most GCMs, although nudged to reanalysis, do not reproduce the seasonal wind pattern at the MLT nor do some models show the correct signs in the zonal mean winds compared to the observations presented here in. From linear theory, it is well-known that a biased mean zonal wind affects also the wave propagation and phase behavior. In so far, we kept a much shorter discussion of the climatologies in section 5.

**Comment:**

3. (L8-12 on p.7, L 3 on p.20) For the analysis, the authors assume that the vertical wavelengths of tides are much larger than 25 km. This may be true for semidiurnal tide but is not true for diurnal tide. The wavelength of the gravest propagating mode for the diurnal migrating tide is 25-30 km (the higher modes have shorter wavelengths; e.g., Chapman and Lindzen, 1970) and Davis et al. (2013) (cited by this manuscript) actually reported that the observed wavelengths were sometimes 20-30 km. In this sense, I am wondering whether the vertical retrieval kernel of 16 km damps a part of tides (esp. diurnal tides)? Also, is the averaging kernel applied also to NAVGEM-HA data for comparison?

**Reply:**

The reviewer brings up a good point that we carefully looked into this aspect in previous studies (Baumgarten and Stober, 2019). The ASF technique is not using a fixed vertical wavelength as cut off wavelength nor does the ASF use a vertical average to obtain the tides. The ASF constrains the smoothness of the phase behavior of the tide within the retrieval kernel, which means even a 7 km vertical wavelength tide would be nicely detected as long as the phase would not change dramatically (phase jumps are removed) within the 16 km average kernel.

Further, we have to note that comparison of Davis et al., 2013 was performed for low-latitude observations. A diurnal tidal mode with 7 km vertical wavelength is basically not observed at mid- and high-latitudes. Climatologies of the diurnal tide suggest that the amplitudes at mid- and high latitudes are much smaller compared to the semidiurnal tide and peak at around 100 km during the whole year, which is outside the altitude range of NAVGEM-HA so far. There is also a weak secondary maximum in the summer mesopause visible, but not of relevance for the study here.

**Comment:**

4. Section 5.4 might be potentially interesting, but the main conclusions are not clear for the present manuscript. For example, the authors first say that the semidiurnal amplitude is enhanced after SSW (L3 on p.14) and the observed phase is approaching that of Pekeris resonance (~M2?) (L21-22 on p.24), but later said (L1 on p.26) that there was no amplification of lunar tide; these descriptions seem contradicting and confusing. At the final paragraph (L2 on p.27) it is suddenly suggested that the day-to-day variability is attributed to zonal winds, but this seems just a speculation without any evidences. At the beginning of the discussion (L21 on p.23), the authors say "we want to disentangle these three aspects…". I wonders what is the final conclusion in this context?

**Reply:**

The holographic analysis shows that the semidiurnal tide shows frequently a drift towards the 12.4 h period, which should fall into the Pekeris resonance according to Zhang and Forbes (2014) (see also reply below). Although the phase variability is drifting towards the period of the lunar tide no enhancement is visible in some cases. This becomes even more obvious when comparing the hologram to the lunar orbit. We looked into the lunar distance, the azimuth and elevation angle relative to our local and global references and found no correlation to our tidal analysis. There is no obvious connection between a phase drift of the semidiurnal tide and the lunar orbit. Now, one could argue that only during the SSW the thermal and dynamic structure satisfies the resonance condition as presented in the theory about the Pekeris resonance (see reference in Forbes and Zhang (2012)). However, our analysis shows that the enhancement after the SSW last longer (approx. 10 days) than the SSW itself (4 days) and there is a time delay of several days (3-5 days). So there is a mismatch of the duration of the enhancement and the time period that satisfies the resonance condition. Given that the ASF technique represents a more realistic true interday variability due to the short wind window length compared to many of the previous analysis, which were based on 21/24-day window analysis. Further, considering that the spectral line shape obtained from the global analysis remains symmetric, although the hologram shows that SW2 phase shift that approaches the 12.42 h period, but no shoulder or enhancement in the spectral domain becomes visible.

**Comments:**

5. For the Pekeris resonance, how is the resonance period determined? The resonance period depends on the background zonal wind and temperature each time. Forbes and Zhang (2012) examined the case for the January 2009. The present study considers the cases in the years 2010 and 2013, and so Forbes & Zhang's results cannot be simply applied.

**Reply:**

In fact, the reviewer made a good point only Forbes and Zhang (2012) actually computed the shift of the Pekeris resonance explicitly using the GSWM model for the SSW 2009. However, later **Zhang and Forbes (2014) argued somehow that we quote as it is difficult to express it much better. "**During the typical SSW years, e.g., 2006, 2009, or 2013, the $M2$ responses are significant with peak amplitudes at 25–30 K. Utilizing the Global-Scale Wave Model, *Forbes and Zhang* [2012] showed that $M2$ amplification during 2009 was due to a shifting of the Pekeris peak to $M2$'s oscillation period at 12.42 h. It is reasonable to infer that during January of 2006 or 2013, the Pekeris frequency peak was also very close to 12.42 and with greater magnitude. The interesting thing is that almost every year, the $M2$ lunar tide has some degree of amplification that peaks at some day during January and February and lasts for 15–20 days. This is not surprising if we recall that the Pekeris peak is not very sharp but instead somewhat broad. Even if the Pekeris peak is not exactly at 12.42 h, part of its shape may also coincide with 12.42 h so that $M2$ gets amplified."

Although, we did not explicitly compute the resonance condition, the holographic analysis provides some indication of the resonance period, which is more or less defined by the time span of the red shift after the central day of the SSW. Further, we estimate the semidiurnal tidal wave properties with respect to the mean frequency and vertical wavelength (see appendix of revised manuscript). It turns out that before the SSW and during the enhancement a semidiurnal tide with about 50-60 km vertical wavelength is present. Only during the second phase of the SSW (after the central day), the mean frequency indicate a lunar period of about 12.42 h and a vertical wavelength of 200-400 km. However, it needs to be confirmed with models like the GSWM whether these changes point towards

a lunar tide, or whether they are the result of the superposition of the non-migration, migrating and lunar tide. However, due to the much better temporal resolution of the ASF in combination with the holographic analysis such very transient and intermittent characteristics of the tidal variability can be studied.

**Minor comments:**

**Comment:**
1. L34 on p.6: What do you mean by "considering also a potential semidiurnal and terdiurnal tide"?

**Reply:**

We removed the word 'potential'. As it is not needed, although the amplitude can become effectively zero.

**Comment:**
2. L1 on p.7: What do you mean by "regularization"? Do you mean that semidiurnal tide is fitted to the residual time series after removing the daily-mean and diurnal tide?

**Reply:**

Regularization is a mathematical expression to constrain fits adding other properties to the derived quantities or a priori knowledge. The most common regularization is the Tikhonov regularization for L2-norm functions. L1-norm regularizations are called Lasso-type.

Here we perform a regularization of the tidal waves considering large scales for the smaller windows. A classical harmonic fit of mean winds and the diurnal, semidiurnal and terdiurnal tide has 7 free parameters. Firstly, we use a window with more than 24 hours, which contains at least 24 measurements to fit the mean winds and diurnal tide. These values are than used as regularization for the smaller window adapted to the semidiurnal and terdiurnal tide. This shorter window contains only 12 points, however, as we have already knowledge of the mean wind and diurnal tide we don't need to fit again for these parameters, but we can use them as boundary for the semidiurnal tidal fit and regularize the new fit. The result is that we have only 4 unknown in the adapted shorter window and we avoid an explicit computation of the residuals. Mathematically, we solve the Jacobien matrix as block diagonals using a Tikhonov regularization for the already determined parameters.

The algorithm is still under development and further releases are planned to include other type of constraints. The vertical regularization is implemented as second iteration step after we solved in a first guess using the temporal only ASF. Mainly to avoid to intense use of computational resources, as the matrices get soon large and more complicated.

**Comment:**

3. L10 on p.7: For the "16 km vertical retrieval kernel", does this averaging(?) kernel put the same weights for the 16 km range? What kind of waves (wavelengths) would be effectively filtered with this kernel? (please also see Major comment #3).

**Reply:**

We apply a full error propagation of the statistical raw voltage fluctuations from the antenna expressed by a radial velocity error until the finally derived wind or tidal parameter. The retrieval is in more detail described in Stober et al., 2018 and Gudadze et al., 2019.  All measurements within the retrieval kernel are weighted by their statistical uncertainty.

**Comment:**

4. L8 on p.11: It is said "the latter point (minimum in Nov.?) is not visible above Tavistock…" but for me, November minimum is clear at Tavistock.

**Reply:**

We rephrased the sentence. The comment only refered to the April semidiurnal tidal enhancement, which is visible at Tavistock, but is absent at Juliusruh and Andenes.

**Comment:**

5. L10 on p.12 "due to the coarser temporal resolution of the global data": How can you reach this conclusion?

**Reply:**

We removed this sentence. The coarser temporal resolution is not dramatically impacting the phase and amplitude variations at seasonal scales. This is not of relevance for the conclusions.

**Comment:**

6. L10-11 on p.26: How are these two peaks explained vortex splitting event and planetary wave activity? Please describe more details.

**Reply:**

We expanded the discussion of the planetary wave and it affects the local measurements.

**Comment:**

7. L21-24 on p.28: I feel that this is just a speculation.

**Reply:**

We rephrase the statement. Tidal mean flow and mean flow tidal interactions are commonly discussed in the literature. Considering that tides also hold the polarization and dispersion relation of gravity waves for a dissipation free atmosphere based on the primitive equations (She et al., 2016), and, thus, the phase variability is explainable by changes in the mean flow. The seasonality of the mean zonal winds is also reflected by a seasonality of the phase behavior of the tide. In particular, the transitions times from winter to summer and from summer to winter including the asymmetry is reflected in the morphology of the phases. Apparently, the semidiurnal tide gains only significant amplitudes during eastward zonal winds and is less present during the westward wind phase.

Further, the tide seems to show an enhancement after a reversal from westward winds to eastward winds at the mesosphere.

However, in particular, the response of the semidiurnal tide on a global and local scale to SSWs seems to provide convincing evident that the wind and the thermal structure are highly relevant for the tidal structure at the MLT. From this perspective the statement seems to be not very speculative.

**Comment:**
8. Captions of Fig.3-12…: Please do not repeat the same sentences in captions.

**Reply:**

We replaced repeating captions with same as but for …

[revised manuscript text omitted]

---

## Author Response (AR3)

**General reply:**

We thank the reviewer for his constructive feedback on our manuscript. This review was very much appreciated as helped a lot to see aspects that are not obvious when working too much on the data analysis. It would be good, if every review would be done with such care and detailed background knowledge.

We revised the manuscript and included the suggestions of this review. A detailed response to the points is found below.

**Major comment** (Pages and Line numbers are based on the manuscript with track of changes):

1. Although the present manuscript is much easier to follow compared with the previous one, I still cannot get the point of the very long discussion in Section 5.3. The authors say at Line 11 of Page 26 that "We discuss these three aspects using….". I assume that these three aspects include (1) changes in mean wind, (2) nonmigrating tides and (3) lunar tidal amplification. At this point, readers expect the following paragraphs to discuss each of them; but the actual discussion is not straightforward (not well organized) but winding. For instance, another factor (i.e., ozone change) suddenly appears; I cannot understand how the discussion of vertical wavelengths is relevant here. While reading the answers to my previous major comment #4, I got the impression that the authors try to exclude the possibility of (2) and (3), and thus to speculate that (1) may be important (Is it true?). Since the suggestion that (3) is not the case seems important, please consider again making the discussion of this Section as concise as possible, really focusing on the discussion of each of (1)-(3). Otherwise readers may get lost and have difficulty getting to the final conclusion.

**Reply:**

We sorted the paragraphs improving the readability of the discussion section. This required some changes in the wording of the paragraphs to make them fit better together. We also rephrased the introduction sentence of the discussion to mention the lunar orbit and holographic analysis.

The vertical wavelength analysis is another key feature to rule out that the lunar tide leads to the amplification. This is the first time that all 4D parameters of a tidal wave are determined from global and local diagnostic, which means we can describe the k-wave vector and the period of the tidal wave and their temporal evolution.

Considering the frequency, time line and zonal and vertical wavenumbers the tidal enhancement after the SSW is truly not driven by the lunar tide or the coupling has to be much more complicate as suggested from the Pekeris resonance theory.

Other comments

**Comment:**
1. (L7-8, p24) "higher temperature in NAVGEM-HA may also explain the higher wind magnitudes" I could not understand the logic.

**Reply:**

NAVGEM-HA winds in the mesosphere are based in part on assimilated temperatures that provide a small adjustment (or increment) to the background state via the thermal wind balance (thermal wind equation). Thus, horizontal grdients in the assimilated mesospheric temperature field help determine vertical gradients in mesospheric winds and their corresponding magnitude. A warmer mesosphere with stronger gradients compared to the 'true' atmospheric state could therefore result in increased wind magnitudes. Other factors such as model physics (e.g., parameterized gravity wave drag) may also contribute to the larger wind amplitudes near the model top. Further study is needed to better understand the origins of this disagreement and improve NAVGEM-HA performance in the future.

**Comment:**
2. (L11, p25) "?" Something is missing.

**Reply:**

There was a typo in the citation statement. This is corrected.

**Comment:**
3. (Eqs (3-6)) The amplitude change with time does not need to be considered?

**Reply:**

Amplitude modulation does not require a change in phase. This was a classical radio technique for communication with a fixed carrier frequency. The hologram as such just uses the phase information, the amplitude modulation adds the intensity, but is not relevant for the spectral variability or Doppler shifts. In remote sensing very often the phase can be measured more precisely and is more accurate than the amplitude.

On the other side the amplitude modulation is very interesting to understand the geophysical reason for the modulation. At the MLT it could be the result of the source variability or propagation of the waves. However, this goes beyond the scope of this paper, but will be part of future studies. In particular, the ozone concentration and layering are important sources for this variability.

**Comment:**
4. (Fig. 14) How long data are used for drawing Fig.14 (only during 2012/2013 SSW?; please describe such information in caption).

**Reply:**

We added this information in the caption as suggested.

**Comment:**
5. (L4 p.31) "Instrumental effects" How can you conclude this? For me the secondary peak appears close to the M2 frequency.

**Reply:**

The meteor radar at Juliusruh did suffer in the season winter season 2012/13 from several technical problems, which appears to be less dramatic due to the performance of the ASF. We double checked the side peaks by comparing to other years and systems. Therefore we investigated the Collm MR data, which is located 300 km south of Juliusruh. Please see attached the corresponding spectral lines obtained from the holograms.

spectral line shape semidiurnal tide MR Collm

[Figure]

**Comment:**

6. (Appendix) Figures should be cited in the text?

Done.

[revised manuscript text omitted]

---

## Author Response (AR4)

Dear William,

technical correction to "Comparative study between ground-based observations and NAVGEM-HA analysis data in the MLT region"

In your decision statement you raised a new concern about the terminology that is used in the manuscript. The suggestion was to replace holographic analysis by hodogram or hodographic.

When the manuscript was written, we searched in the literature to find an appropriate terminology of what is done. A simple look to Wikipedia provides already a short summary on the terms hologram and hodograph and its main features.

The definition of a hologram that is currently available in Englich Wiki shows the classical laser hologram. However, a hologram is mathematically a superposition of a coherent wave with an imaging wave that is diffracted or interfered with an object. Storing the information of the phase differences into an image matrix allows to reconstruct the 3D structure of that object from this 2D image. This can be done in transmission, reflection as phase-only or mixed type of holograms.

Our reference wave is the excitation of tides at the troposphere/stratosphere with the solar orbit precision. This excitation is highly coherent. Atmospheric tides then propagate vertically through the atmosphere. The meteor radar just takes picture of the transmitted wave field, or the through the atmosphere transmitted tide. All interactions with the mean wind, planetary waves, gravity causes lead to small changes of the phase and amplitude. Thus, we infer these tiny changes in phase of the coherently excited wave by the holographic analysis. Mathematically, this is done as outlined in the manuscript. We truncated the Taylor expansion already after the first term, however, if we include the second order term, we obtain the Fresnel transform or 1D-holography introduced by Elford (2004).

A hodogram or hodograph is essentially different in that respect as the phase plays a minor role. It is the vector length (magnitude) that is added and there is a clear dependency from the starting point. A hodograph is basically a dead reckoning navigation or ray-tracing of information. In this case our analysis would dependent on the starting point. So the hodograph just containing the January and February would look different from the one that started in December. This sensitivity of the hodograph analysis is well-know from gravity wave analysis. It can be very challenging to derive the gravity wave properties from a hodograph analysis, as small changes in the starting point already lead to quite some changes in the final result. This is not the case in our analysis. The results do not depend on whether we just look at the January or at the complete winter season.

$x(i+1) = x(i) + f'(x)$

Further, the holographic analysis is phase only and does not depend on the amplitude of the tidal wave.

Thus, we suggest keeping the terminology as it is. However, we added some sentence to the paragraph to point out that we used a more abstract definition of the holography. The added sentence is labelled in bold. The text basically explain of what we wrote in this report.

In the following, we investigate the phase variability of the semidiurnal tide introducing a holographic analysis for the SSW 2012/13 and discuss a potential connection to the Pekeris resonance \citep{Zhang_Forbes_2014_lunar_tide_Pekeris}. **Similar to other holographic analysis, we use the phase differences between a coherent reference wave and the observed wave field to infer small deviations in frequency that are not resolvable by standard Fourier techniques.** The day-to-day variability, obtained from the ASF, indicates that the tidal phase are not stable with time and show significant interday variability, which appears to be related to changes in the zonal wind in the middle atmosphere driven by the polar vortex and planetary waves. Considering that a time dependent phase corresponds to a frequency shift, it is possible to convert this temporal phase variability into a period change and, hence, to estimate the spectral line shape of the tide or to derive a holographic representation of the temporal evolution on a day to day basis.

Please let us know, whether these explanations are satisfying the concerns that were raised.

Best regards,

Gunter